# A *KRAS*-responsive long non-coding RNA controls microRNA processing

Lei Shi [1,2], Peter Magee[1,2], Matteo Fassan [3], Sudhakar Sahoo[4], Hui Sun Leong[4], Dave Lee[4], Robert Sellers[4], Laura Brullé-Soumaré [5], Stefano Cairo [5], Tiziana Monteverde[1,2], Stefano Volinia[6], Duncan D. Smith[7], Gianpiero Di Leva[8], Francesca Galuppini[3], Athanasios R. Paliouras[1,2], Kang Zeng[9], Raymond O'Keefe [10] & Michela Garofalo [1,2 ✉]

Wild-type *KRAS* (*KRAS*^WT) amplification has been shown to be a secondary means of KRAS activation in cancer and associated with poor survival. Nevertheless, the precise role of *KRAS*^WT overexpression in lung cancer progression is largely unexplored. Here, we identify and characterize a KRAS-responsive lncRNA, *KIMAT1* (ENSG00000228709) and show that it correlates with KRAS levels both in cell lines and in lung cancer specimens. Mechanistically, *KIMAT1* is a MYC target and drives lung tumorigenesis by promoting the processing of oncogenic microRNAs (miRNAs) through DHX9 and NPM1 stabilization while halting the biogenesis of miRNAs with tumor suppressor function via MYC-dependent silencing of p21, a component of the Microprocessor Complex. *KIMAT1* knockdown suppresses not only KRAS expression but also KRAS downstream signaling, thereby arresting lung cancer growth in vitro and in vivo. Taken together, this study uncovers a role for *KIMAT1* in maintaining a positive feedback loop that sustains KRAS signaling during lung cancer progression and provides a proof of principle that interfering with *KIMAT1* could be a strategy to hamper KRAS-induced tumorigenesis.

[1] Transcriptional Networks in Lung Cancer Group, Cancer Research UK Manchester Institute, University of Manchester, Manchester, UK. [2] Cancer Research UK Lung Cancer Centre of Excellence, at Manchester and University College London, London, UK. [3] Department of Medicine, Surgical Pathology & Cytopathology Unit, University of Padua, Padua, Italy. [4] Computational Biology Support, Cancer Research UK Manchester Institute, University of Manchester, Manchester, UK. [5] Xentech, 4 rue Pierre Fontaine, Evry, France. [6] Department of Morphology, Surgery and Experimental Medicine, University of Ferrara, Ferrara, Italy. [7] Biological Mass Spectrometry Facility, Cancer Research UK Manchester Institute, University of Manchester, Manchester, UK. [8] School of Pharmacy and Bioengineering, Keele University, Stock-on-Trent, UK. [9] Imaging & Cytometry Facility, Cancer Research UK Manchester Institute, University of Manchester, Manchester, UK. [10] Division of Evolution & Genomic Sciences, School of Biological Sciences, Faculty of Biology, Medicine and Health, University of Manchester, Manchester, UK. ✉email: michelagarofalo1@gmail.com

While drugs that inhibit KRAS for therapeutic gain begin to be available in the clinic for patients with specific KRAS mutations[1], the functional role of $KRAS^{WT}$ amplification, which has recently been shown to be a secondary means of KRAS activation in cancer[2], remains mostly unexplored. The largest part of the human genome is composed of non-coding genes, which are transcribed but not translated into proteins, defined as non-coding RNAs. The most abundant class of these non-coding transcripts in mammals is represented by long non-coding RNAs (lncRNAs), RNAs molecules longer than 200 nucleotides which affect gene expression through interaction with DNA, RNA or proteins[3]. High-throughput RNA sequencing studies have led to the identification of lncRNAs with oncogenic or tumor suppressive properties in many cancer types, including lung cancer[4–6]. Although lncRNAs are clearly a crucial layer of biological regulation, only a few have been functionally characterized and the molecular mechanism(s) behind their biology remains largely unknown.

Another class of regulatory non-coding RNAs, miRNAs, have gained significant importance in the past decades as master regulators of gene expression[7]. MiRNAs bind to the 3'UTR of target genes inhibiting their translation or inducing mRNA degradation. MiRNA biogenesis is a complex process in which the primary transcripts of miRNA genes (pri-miRNAs) are cleaved in the nucleus into precursor miRNAs (pre-miRNAs) by the Microprocessor Complex (MC), essentially composed of the RNase III DROSHA, DGCR8 and RNA associated proteins including p68 (DDX5) and p72 (DDX17)[8–10]. Pre-miRNAs are further processed in the cytoplasm into mature miRNAs by Dicer, another RNase III related enzyme. MiRNA dysregulation is a hallmark of human cancers and miRNA roles as tumor promoters (TP) or tumor suppressors (TS) have been validated in several studies to date[11]. Here, we leverage the pleiotropic functions of lncRNAs to potentially identify pathways regulated by KRAS that could be exploited for therapeutic purpose. We characterize a human KRAS-responsive lncRNA, KIMAT1, which sustains essential oncogenic signaling by modulating components of the MC and miRNA biogenesis, thereby promoting lung cancer progression.

We have therefore, uncovered a function for a lncRNA in miRNA processing, which may shed light on key aspects of cancer biology.

## Results

**Identification of KRAS-responsive lncRNAs.** Through in silico analysis of KRAS copy number alteration (CNA) in human clinical samples from the Cancer Genome Atlas (TCGA), we identified high level amplification of the KRAS gene, as previously reported[12], as well KRAS copy number gain (see methods) in both lung adenocarcinoma (LUAD) and lung squamous cell carcinoma (LUSC), with consequent increase of KRAS mRNA (Fig. 1a and Supplementary Fig. 1a). 17% of LUAD patients with KRAS gain/amplification also harbored a mutant KRAS allele (Supplementary Data 1). Kaplan–Meier survival analysis revealed that patients with amplified KRAS had a poorer disease-free survival compared to patients with nonamplified KRAS status (Fig. 1b). To identify potential KRAS-modulated pathways we searched for KRAS-responsive lncRNAs. We carried out RNA sequencing (RNA-seq) analysis after overexpression (OE) of either $KRAS^{WT}$ or $KRAS^{G12D}$ in H1299 cells, which although harboring an NRAS mutation do not depend on NRAS signaling[13]. Setting a threshold fold change (FC) > 1.5 or <0.8 and padj < 0.05, a total of 1783 protein-coding genes and 104 lncRNAs were concordantly induced or repressed by $KRAS^{WT}$ or $KRAS^{G12D}$ (Supplementary Fig. 1b) (raw data accessible via GSE124631), suggesting that a substantial portion

of the KRAS signaling remains uncharacterized. Gene set enrichment analysis (GSEA) revealed that "MYC targets", "angiogenesis" and "epithelial to mesenchymal transition (EMT)" signatures were significantly induced in the KRAS associated genes (Supplementary Fig. 1c and Supplementary Data 2). The top-scored lncRNAs induced by KRAS, with a fold change above 3, were Linc02575 and HIF1A-As2 (Fig. 1c). While HIF1A-as2 has been previously reported to have an oncogenic role in different tumor types[14,15], Linc02575 has never been characterized before. Thus, in this study we focused on Linc02575, thereafter referred as KIMAT1 (KRAS-Induced-Metastasis-Associated-Transcript 1). KIMAT1 is a long intergenic non-coding RNA located on chromosome 21 with only one isoform (Supplementary Fig. 1d). KIMAT1's transcript full length (912 nt) was determined by a rapid amplification of cDNA ends (RACE) (Supplementary Fig. 1e). KIMAT1 is not conserved in other species (Supplementary Fig. 1f) and its secondary structure is shown in Supplementary Fig. 1g. Coding Potential Assessment Tool (CPAT)[16] with KRAS and MALAT1 as controls, was used to validate that KIMAT1 is a non-coding transcript (Supplementary Fig. 1h). We next verified regulation of KIMAT1 by $KRAS^{WT}$ or $KRAS^{G12D}$ in multiple cell lines, including the normal immortalized BEAS2B cells which harbor $KRAS^{WT}$, confirming that KRAS OE increased whilst KRAS silencing decreased KIMAT1 expression (Supplementary Fig. 1i–k). Additionally, silencing or inhibition of molecules upstream or downstream of KRAS led to KIMAT1 downregulation (Supplementary Fig. 1l, m). To decipher the contribution of KIMAT1 in KRAS-induced tumorigenesis, we generated BEAS2B and H1299 cells stably overexpressing $KRAS^{WT}$. In both cell lines $KRAS^{WT}$ OE increased cell proliferation and 3D cell invasion with a rescue of the phenotype upon KIMAT1 knockdown (KD), evidencing that KIMAT1 is a crucial mediator of KRAS-induced tumorigenesis (Supplementary Fig. 2a–d). Next, to verify whether KIMAT1 could be clinically relevant, we identified genes that were both upregulated upon KRAS OE in H1299 cells and in the LUAD and LUSC datasets from the TCGA compared to normal lung tissues. Interestingly, KRAS was among the most differentially expressed protein-coding genes, whilst KIMAT1 was among the top expressed KRAS-modulated lncRNAs in lung cancer specimens (Fig. 1d). In situ analysis of two additional independent cohorts confirmed that adenocarcinoma and squamous cell carcinoma lesions expressed higher levels of KRAS and KIMAT1 compared to the corresponding normal counterpart with a strong positive correlation (Fig. 1e, f and Supplementary Fig. 3a–c). Notably, there was a significant increase in KRAS and KIMAT1 expression in late stage compared to early stage adenocarcinoma lesions, suggesting that KIMAT1 expression increased progressively in proportion to KRAS levels (Fig. 1e and Supplementary Fig. 3d). KIMAT1 was also overexpressed in cells with high KRAS copy number (Supplementary Fig. 3e), confirming a direct correlation between KIMAT1 and KRAS also in vitro. In support of an oncogenic role, we detected KIMAT1 expression in several other tumor types and cancer cell lines with low or no expression in normal tissues (Supplementary Fig. 3f).

**KIMAT1 originates from Transposable Elements (TEs) and is activated by MYC.** To understand the KIMAT1 mechanism of regulation we first identified the 5' Transcription Starting Site (TSS) by Cap Analysis of Gene Expression sequencing (CAGE-seq) (Fig. 2a). In-house ChIP-seq data from H1299 cells revealed peaks of H3K4me3 and H3K27ac at the KIMAT1 promoter region, underlying the presence of an active promoter and "open chromatin" (Fig. 2b). Then, we used JASPAR

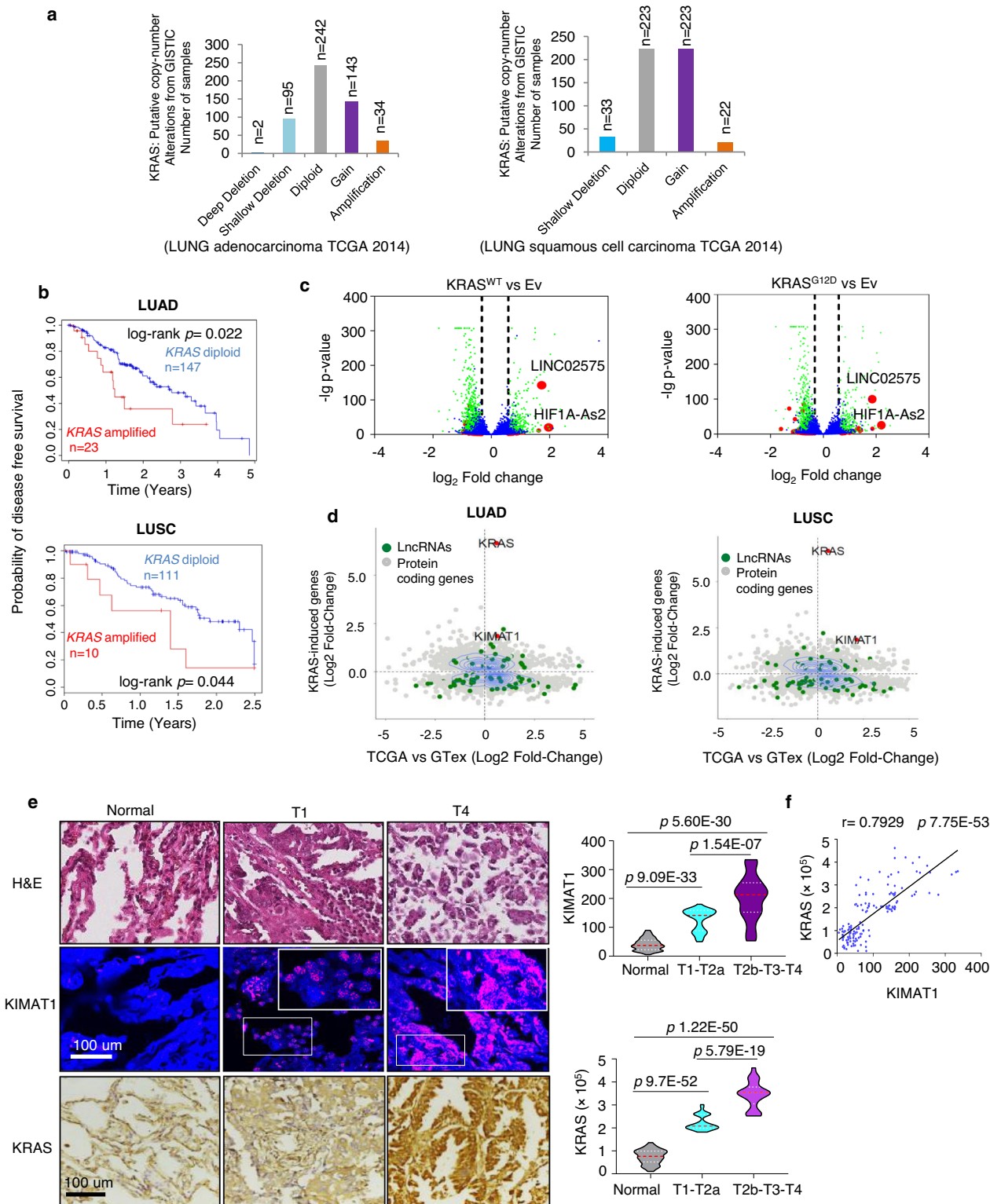

(http://jaspar.genereg.net), an open-access database of matrix-based nucleotide profiles, to determine binding preference of transcription factors[17]. This analysis identified binding sites for MYC, fundamental in KRAS-induced transformation, in the *KIMAT1* promoter[18]. Previously published ChIP-seq data from A549 cells revealed increased level of MYC-binding proximal to the TSS and overlapping with H3K4me3 and H3K27ac peaks, as expected in the case of a direct binding (Fig. 2b). Almost 50% of the human genome sequence contains interspersed repetitive

sequences originated from mobile transposable elements (TEs), which have been recently reported to be involved in lncRNA origin and regulation[19]. TEs insert into the genome regulatory sequences, including transcription factor binding sites[20]. For instance, subfamilies of TE long terminal repeats (LTRs) harbor functional MYC response elements responsible for transcriptional activation[20]. Interestingly, inspection of the *KIMAT1* promoter revealed that two MER101 LTRs, derived from endogenous retrovirus 1 (ERV1) sequences located at 775 bp and 359

**Fig. 1 *KRAS*-responsive lncRNAs. a** KRAS putative copy number alterations from GISTIC in LUAD and LUSC (TCGA), including homozygous deletions (deep deletion, CN = −2), heterozygous deletion (shallow deletion, CN = −1), normal diploid (CN = 0), gain (CN = 1) and amplification (CN = 2 or more). LUAD total number of patients = 513. LUSC total number of patients = 502. **b**, Kaplan–Meier survival curves comparing probability of disease free survival of patients with high *KRAS* expression or amplified KRAS (red line) to patients with low *KRAS* expression or nonamplified KRAS (blue line) from the TCGA datasets LUAD and LUSC. The log rank *p* values were obtained from a two-tailed Chi-Square test. **c** Volcano plots of significant KRAS-responsive lncRNAs determined by RNA-seq in H1299 cells overexpressing either KRAS$^{WT}$ or KRAS$^{G12D}$ compared to cells transfected with an empty vector (Ev) (padj < 0.05). **d** Identification of *KRAS*-regulated genes and lncRNAs in LUAD and LUSC lesions. Scatterplots indicate expression in LUAD and LUSC of KRAS-regulated genes identified in Fig. 1c. The y-axis shows the log$_2$-transformed fold change in gene expression following KRAS OE, the x-axis depicts the log$_2$-transformed difference in gene expression between lung adenocarcinoma (n = 540) and normal lung (GTex, n = 427) and between squamous cell carcinoma lesions (n = 501) and normal lung (GTex, n = 427). **e** (Left) Representative images of *KIMAT1* and KRAS in matched FFPE normal-tumor adenocarcinoma lesions at stage 1 (T1) and stage 4 (T4) stained with DAPI (blue), *KIMAT1* (pink) (smFISH) or KRAS (brown). Scale bar, 100 μm. (Right) Quantification of *KIMAT1* and KRAS expression by smFISH in tumor and normal lung tissues reported in e, Spots were counted using the online JAVA software from StarSearch. **f** Direct correlation between *KIMAT1* with KRAS in adenocarcinoma lesions reported in e by Pearson's rank correlation coefficient (*r*). **e, f** LUAD normal n = 75, T1 = 37, T2a = 14, T2b = 4, T3 = 16, T4 = 4. Error bars indicate mean ± S.D. *p* values were calculated by two tailed Student's *t* test.

bp upstream *KIMAT1* TSS contained two and one MYC BS, respectively (Fig. 2b, c).

Furthermore, the entire *KIMAT1* genomic locus was essentially composed by remnants of ERV1 and L1 DNA in both exons and introns. To verify whether the MER101 LTRs had potential promoter activity, we cloned their sequences in a promoterless pGL3 Basic vector. Dual luciferase assay revealed that the LTR located 775 bp upstream of the TSS (MER-101-1) and containing two MYC-binding sites (Fig. 2c), had promoter function, as the luciferase expression increased compared to control (Fig. 2d). As expected, MYC silencing reduced the expression of the reporter gene fused to the MER-101-1 sequence with a rescue of the luciferase activity upon deletion of the two MYC-binding sites. A non-promoter region (NP) was used as negative control (Fig. 2d). MYC binding to *KIMAT1* promoter was further corroborated by ChIP-qPCR (Fig. 2e) while MYC silencing reduced *KIMAT1* expression (Fig. 2f). In a rescue experiment, MYC KD abolished the induction of *KIMAT1* by KRAS (Fig. 2g). Together, these findings suggest that *KIMAT1* is a human lncRNA originated by evolutionary ERV1 integration events in the human genome and transcriptionally activated by MYC.

**KIMAT1 is essential for cancer cell survival, growth and invasion.** Single-molecule RNA FISH (smFISH) and subcellular fractionation experiments indicated that *KIMAT1* is localized both in the cytoplasm and in the nucleus (Fig. 3a, b). To elucidate *KIMAT1* biological functions we designed three different LNA-GapmeR (GpR) antisense oligonucleotides (ASOs) to silence *KIMAT1* expression (Supplementary Table 1). GpRs transfection resulted in marked *KIMAT1* KD efficiency as assessed by qPCR and smFISH (Supplementary Fig. 4a, b). *KIMAT1* silencing dramatically reduced cell proliferation, 3D cell invasion and clonogenic ability in several cancer cell lines and induced remarkable cell death (Fig. 3c–e and Supplementary Fig. 4c–e). The same effects were observed upon *KIMAT1* silencing using two independent siRNAs (Supplementary Fig. 4f, g). Reciprocally, cells with stable lentiviral *KIMAT1* OE (Supplementary Fig. 5a) exhibited a consistent increase in 3D cell invasion, growth in NOD/SCID gamma (NSG) mice and long-term survival in clonogenic assays (Fig. 3f–h and Supplementary Fig. 5a–c). To further verify the specificity of the GpRs and exclude "off-targets" effects, we deleted the region in *KIMAT1* necessary for the GpRs binding (Supplementary Fig. 5d). While, cells transfected with a plasmid harboring *KIMAT1* WT showed reduced 3D cell invasion and colony formation upon GpRs transfection, cells overexpressing mutant *KIMAT1* lacking the GpR targeting site remained unaffected, suggesting that the biological effects observed are *KIMAT1* specific (Supplementary Fig. 5e, f).

Furthermore, mutation in the GpR targeting site significantly reduced the spheroid area in a 3D invasion assay, evidencing that the deleted region could be a functional element in the lncRNA sequence.

**KIMAT1 interacts with DHX9 and NPM1 and is essential for their stability.** Next, to elucidate the molecular mechanisms through which *KIMAT1* could affect lung cancer progression, we purified endogenous *KIMAT1* RNA complexes by RNA antisense purification coupled with mass spectrometry (RAP–MS)[21] using biotinylated *KIMAT1* RNA antisense probes (Fig. 4a and Supplementary Table 2). Forty-five proteins were reproducibly identified in two biological replicates to interact with *KIMAT1* (Supplementary Data 3). The majority of the retrieved proteins are involved in translation, metabolism or are components of the plasma membrane. Among the enriched proteins, two RNA-binding proteins, DHX9 and NPM1, attracted our attention for their roles in cancer progression[22,23]. RNA pull-down confirmed the binding between *KIMAT1* and DHX9, and between *KIMAT1 and* NPM1 (Fig. 4b and Supplementary Fig. 6a). Lysate pretreated with RNase A or pull down with the *ubiquitin C* (*UBC*) probe were used as negative controls. To verify the existence of a direct binding between *KIMAT1* and DHX9 and between *KIMAT1* and NPM1, we carried out a cross-link RNA immunoprecipitation (CLIP) assay, which induces irreversible covalent bonds between directly interacting proteins and RNA, as previously reported[24]. DHX9 or NPM1 antibody-bound complexes exhibited a significant enrichment in *KIMAT1* (Fig. 4c, d). *Lnc-CCDST* and *SAMD12-As1*, two lncRNAs previously reported to interact with DHX9 and NPM1, respectively, were used as positive controls[25,26]. Sequential immunofluorescence and smFISH evidenced co-localization of *KIMAT1*/DHX9 and *KIMAT1*/NPM1 in both the nucleus and cytoplasm (Fig. 4e, f). To map *KIMAT1* functional motifs that bind to DHX9 or NPM1 we performed deletion-mapping experiments followed by in vitro transcription and pull-down of *KIMAT1* fragments. These experiments revealed that a 399 nt region at the 5' end of *KIMAT1* and a 258 nt region at the 3' end of *KIMAT1* are required for the interaction with DHX9 and NPM1, respectively (Fig. 4g). Two double-stranded RNA-binding domains (dsRBDs) at the N-terminus of DHX9 and a DNA/RNA-binding domain (DRBD) at the C-terminus of NPM1 (Supplementary Fig. 6b) have been previously reported to bind nucleic acids[27,28]. Using DHX9 or NPM1 deletion mutants (Supplementary Fig. 6b) followed by CLIP assay we observed that deletion of the two DHX9 dsRBDs and the NPM1 DRBD domain abolished *KIMAT1* binding (Fig. 4h, i), suggesting that these domains are essential for the binding to *KIMAT1*.

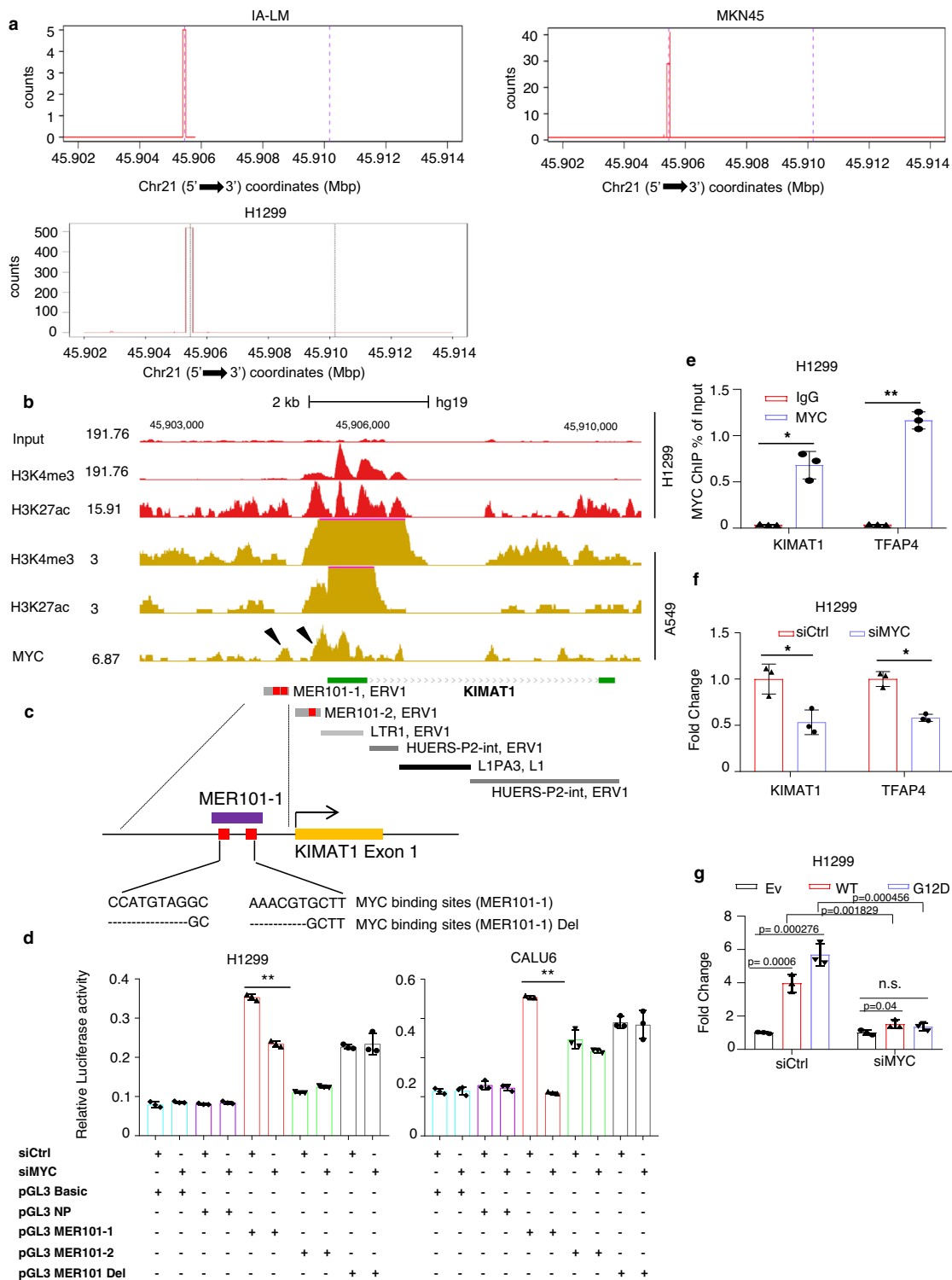

Next, to determine whether deletion of DHX9 or NPM1 binding sites in *KIMAT1* could abrogate its biological effects, we cloned *KIMAT1* full-length and deletion constructs in a lentiviral vector. Overexpression of the mutants gave rise to a lower number of colonies compared to cells transfected with *KIMAT1* full length (Supplementary Fig. 6c, d), revealing that the regions of *KIMAT1* binding to DHX9 or NPM1 are important for *KIMAT1*-mediated cell proliferation and corroborating previous findings (Supplementary Fig. 5e, f). In addition, DHX9 and NPM1 KD using siRNAs or the CRISPR/Cas9 gene-editing system with paired single-guide RNAs (sgRNAs) decreased 3D

cell invasion, 2D and 3D cell proliferation and induced cell death (Fig. 4j, k and Supplementary Fig. 6e–j). Notably, *KIMAT1* depletion decreased whereas *KIMAT1* OE or *KRAS* OE induced DHX9 and NPM1 (Fig. 4l and Supplementary Fig. 7a, b). Therefore, *KIMAT1* is important for DHX9 and NPM1 stability and this effect is attributable to proteasomal degradation as DHX9 and NPM1 levels were rescued upon inhibition of the 26S proteasome complex with the proteasome inhibitor MG132 (Supplementary Fig. 7c) and *KIMAT1* silencing increased the polyubiquitination of DHX9 and NPM1 (Supplementary Fig. 7d).

**Fig. 2 *KIMAT1* originates from TEs and is transcriptionally activated by MYC. a** Cap analysis of gene expression-sequencing (CAGE-seq), from lung and gastric cancer cells revealed the presence of a putative active promoter in *KIMAT1* loci. CAGE-seq counts were defined by the FANTOM5 mammalian promoter expression atlas in the IA-LM and H1299 lung adenocarcinoma and MKN45 gastric cancer cell lines at the *KIMAT1* loci. **b** In-house and University of California, Santa Cruz (UCSC) Genome Browser ChIP-seq data illustrating the recruitment of MYC to the *KIMAT1* loci. Co-localization of H3K4me3 and H3K27ac at the TSS in A549 cells (brown) was integrated with H3K4me3 and H3K27ac in-house ChIP-seq data (red). MYC-binding sites (BS) are indicated by arrows. **c**, Schematic representation of *KIMAT1* genomic locus composed by remnants of ERV1 and L1 DNA and enlargement of the MER-101-1 LTR containing two MYC BSs (in red). **d** Dual-luciferase assay in H1299 and CALU6 cells, showing that the MER101-1 LTR has promoter function as compared to control (Empty vector). MYC silencing reduced luciferase activity, while deletion of the two MYC BSs in the MER-101-1 rescued this effect. A non-promoter region (NP) was used as negative control. **4.12E−05 in H1299 and **3.75E−08 in CALU6. **e** ChIP-qPCR in H1299 cells showing MYC or IgG enrichment (ChIP/input) in the *KIMAT1* promoter region (MER-101-1). TFAP4 was used as positive control. *0.001694, **3E−05. **f** Expression of *KIMAT1* in H1299 cells transfected with control siRNA (siCtrl) or MYC siRNA (siMYC). TFAP4 was used as positive control. siMYC vs siCtrl, *KIMAT1* *0.018211 and siMYC vs siCtrl, TFAP4 *0.001253. **g** MYC KD suppressed the induction of *KIMAT1* by wild type or mutant KRAS. Error bars indicate mean ± S.D, n = 3 replicates. p values were calculated by two tailed Student's t test.

---

### *KIMAT1* and its interacting proteins regulate KRAS signaling.

Analysis of the TCGA datasets LUAD and LUSC and of two independent cohorts of FFPE matched normal/tumor samples revealed that DHX9 and NPM1 are upregulated in lung cancer compared to normal lung and positively correlate with *KRAS and KIMAT1* (Fig. 5a, b and Supplementary Fig. 8a–c). In accordance with a role in promoting tumor progression, high expression of *DHX9/NPM1/KIMAT1* was associated with poor overall survival in lung cancer patients (Fig. 5c). To define the transcriptome modulated by *KIMAT1* and its interacting proteins we performed RNA-seq in H1299 cells transfected with either *KIMAT1*-targeting GpRs or a pool of four different siRNAs targeting DHX9 or NPM1. A total of 6278 (3732 down and 2546 up upon *KIMAT1* KD), 1133 (916 down and 217 up upon *DHX9* KD) and 1324 (1148 down 176 up upon *NPM1* KD) (fold change >1.5 or <0.8, padj <0.05) dysregulated genes were identified (Supplementary Fig. 9a) (data accessible via GSE 124631). A significant overlap between the genes modulated by *KIMAT1* and DHX9 and between the genes modulated by *KIMAT1* and NPM1 was observed (Fig. 5d). Strikingly, we noticed a significant overlap also between genes modulated by DHX9 and NPM1 (Fig. 5d). GSEA analysis revealed that *KIMAT1* KD led to suppression of genes positively regulated by KRAS and enrichment of genes negatively regulated by KRAS (Fig. 5e, f). *KIMAT1* silencing reduced KRAS endogenous levels as well as ERKs and AKT phosphorylation and c-RAF level, supporting the existence of a positive feedback loop (Supplementary Fig. 9b). Among the top enriched gene signatures by GSEA upon DHX9 and NPM1 KD and in common with *KIMAT1* KD were those associated with KRAS, RAF and MEK signaling (Fig. 5f). Randomly selected KRAS target genes were further confirmed by qPCR analysis and immunoblotting (Fig. 5g, h and, Supplementary Fig. 9c, d). In addition, Gene ontology (GO) analysis identified pathways involved in cell motility, cell-cell adhesion and angiogenesis, consistent with the role of DHX9 and NPM1 in tumor progression (Supplementary Fig. 9e). EMT gene signature was enriched in *KIMAT1*, DHX9 and NPM1 KD gene sets (Supplementary Fig. 10a) and selected EMT genes were further confirmed by qPCR and immuno-fluorescence (IF) (Supplementary Fig. 10b–d). In accordance with a role in EMT, cells stably overexpressing *KIMAT1* displayed an elongated phenotype compared to parental cells (Supplementary Fig. 10e). In summary, *KIMAT1* controls oncogenic pathways, including the KRAS pathway, at least in part, through DHX9 and NPM1.

### *KIMAT1* regulates the processing of oncogenic miRNAs through DHX9 and NPM1.

Seeking to define the mechanism through which *KIMAT1* could regulate KRAS expression and global downstream signaling, we noticed that DHX9 had previously been reported to play a role in miRNA biogenesis by interacting with DDX5 (p68) in the MC[9,22]. Because *KIMAT1* stabilizes DHX9, we hypothesized that it might play a role in miRNA processing and therefore regulate gene expression post-transcriptionally. To test this hypothesis, we profiled miRNA expression of *KIMAT1* KD compared to control cells by next-generation sequencing (NGS). Hierarchical clustering analysis identified differentially expressed miRNAs, with 113 miRNAs upregulated and 78 downregulated (Padj < 0.05) (Fig. 6a) (raw data accessible via GSE124631). A majority of the upregulated miRNAs upon *KIMAT1* KD turned out to have tumor suppressor properties whereas most of the downregulated miRNAs have been previously reported to have oncogenic roles[29,30]. For instance, the upregulated miR-200 family members, miR-27a, miR-27b and let-7b upon *KIMAT1* KD are well-known TS miRNAs which play a crucial role in suppressing EMT in many cancer types, including lung tumors[31–35]. On the contrary, among the downregulated miRNAs upon *KIMAT1* loss, miR-17-5p and miR18a, as well as miR-375 and miR-10b have been previously shown to be involved in lung cancer progression[36–39]. Interestingly, network analysis revealed that genes belonging to pathways activated by *KIMAT1*, including the KRAS pathway, were predicted targets of the *KIMAT1*-suppressed miRNAs (Supplementary Fig. 11a and Supplementary Data 4). To exclude that the change in miRNA expression upon *KIMAT1* KD was due to a transcriptional effect we examined the expression of mature, precursor and primary forms of selected *KIMAT1*-modulated miRNAs. While mature and precursor miRNAs were affected by *KIMAT1* loss, the expression of primary miRNAs remained unchanged, implying that *KIMAT1*-mediated miRNA regulation occurs post-transcriptionally (Fig. 6b). In support of this hypothesis, *KIMAT1* KD promoted Drosha-mediated *pri-miR-27b* in vitro processing, performed by incubating radio-labeled *pri-miR-27b* with immunoprecipitated Drosha from H1299 cells (Fig. 6c). Notably, DHX9 KO mimicked *KIMAT1* KD, while DHX9 OE sorted the opposite effect (Supplementary Fig. 11b), suggesting that *KIMAT1* effects on miRNA biogenesis are mediated by DHX9.

To rule out a role for NPM1 in miRNA processing, we analyzed the expression of the *KIMAT1*-modulated miRNAs in NPM1 KO cells or after enforced expression of NPM1. Unexpectedly, we observed a change at the precursor and mature level while the primary remained unaffected, indicating that, as with DHX9, NPM1 participates in miRNA processing (Supplementary Fig. 11c). Since NPM1 is essentially a nuclear protein we first tested whether NPM1 could interact with members of the MC and detected a binding between NPM1 and DDX5 and between DHX9 and NPM1 (Fig. 6d). To substantiate this, we also transfected H1299 cells with a FLAG-tagged DHX9 expression vector and carried out immunoprecipitation. Antibodies against the FLAG epitope precipitated DHX9 along with NPM1 (Fig. 6e),

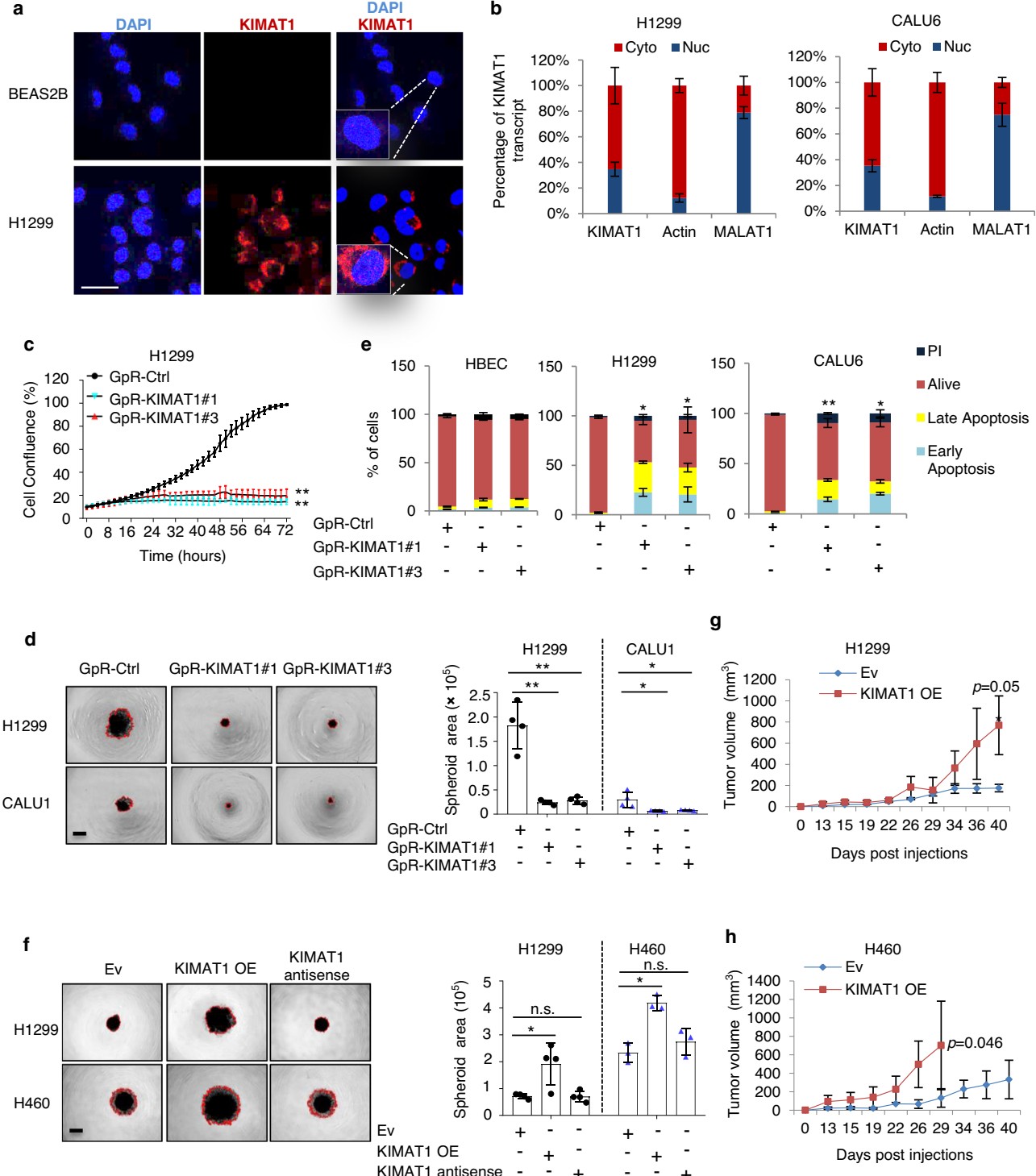

**Fig. 3 *KIMAT1* is an oncogenic lncRNA in NSCLC. a** Confocal microscopy images of *KIMAT1* smFISH (Red) in different cell lines. Images are representative of two biological replicates. Nuclei were stained with DAPI (Blue). Scale bar, 75 μm. **b** qRT-PCR analysis for *KIMAT1* following cytoplasmic (cyto) and nuclear (nuc) fractionation of cell lysates, $n = 3$. **c** Cell proliferation assay upon transfection with two different *KIMAT1* GpRs, n = 4. GpR-KIMAT1#3 vs Ctrl **0.14E−09; GpR-KIMAT1#1 vs Ctrl **1.66E−07. **d** 3D invasion assay of H1299 cells transfected with two different *KIMAT1* GpRs and quantification of the tumorsphere area invading the matrigel, $n = 4$. GpR-KIMAT1#1 vs Ctrl **0.000601 and GpR-KIMAT1#3 vs Ctrl **0.000713 (H1299); GpR-KIMAT1#1 vs Ctrl *0.026964 and GpR-KIMAT1#3 vs Ctrl *0.033348 (CALU1). Scale bar, 500 μm. **e** Quantification of Annexin V staining upon GpRs transfection in multiple cell lines, $n = 3$. *0.013189 and *0.031027 (H1299); **7.99E−05 and *0.020508 (CALU1). **f** 3D invasion assay of H1299-*KIMAT1* and H460-*KIMAT1* stable cells and quantification of the tumorsphere area invading the matrigel. *KIMAT1* antisense was used as control. $n = 4$ (H1299) and $n = 3$ (H460). KIMAT1 OE vs EV *0.022 (H1299); KIMAT1 OE vs EV *0.002105 (H460). Scale bar, 500 μm. **g, h** Tumor growth curves of xenograft mouse models derived from cell lines with stable overexpression of *KIMAT1* ($n = 8$) compared to control mice ($n = 8$). Data show mean ± S.D. *p* values were calculated by two tailed Student's *t* test.

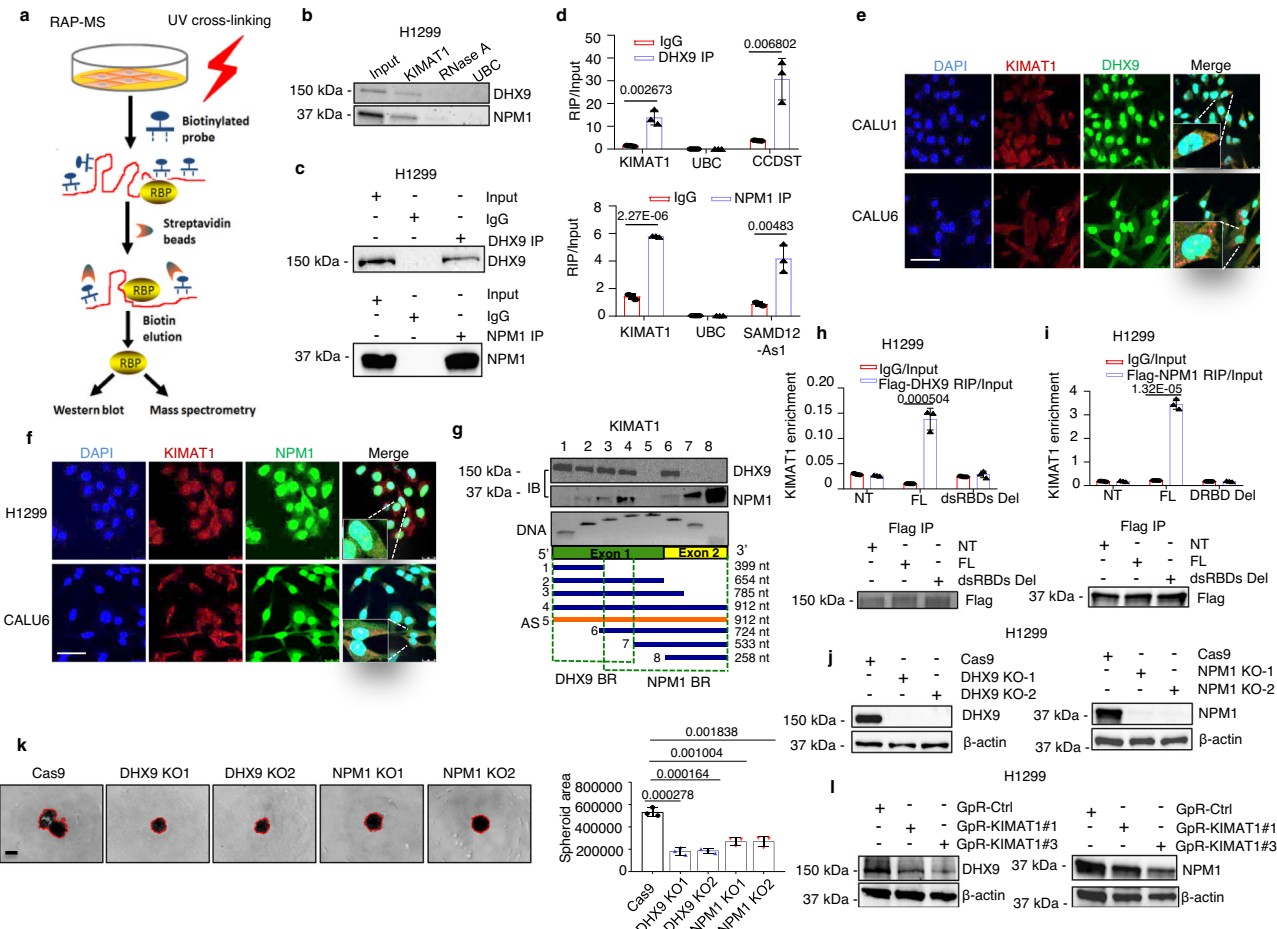

**Fig. 4 KIMAT1 directly interacts with DHX9 and NPM1. a** Schematic representation of RAP-MS. RBP = RNA-binding protein. **b** KIMAT1 pull-down with biotinylated antisense probes after UV crosslinking followed by immunoblotting analysis showing that DHX9 and NPM1 are direct KIMAT1 binding partners. A sample treated with RNase A or pull-down of the housekeeping gene UBC were used as negative controls. **c** Immunoblotting showing DHX9 and NPM1 pull-down efficiency for the experiment in **d**. **d** KIMAT1 is enriched by CLIP assay using DHX9 or NPM1 specific antibodies. **e, f** DHX9 and NPM1 sequential immunofluorescence (green) and KIMAT1 smFISH (Red) in different lung adenocarcinoma cell lines. Scale bar, 75 μm. **g** Schematic representation of human KIMAT1, its antisense (AS) and various deletion constructs generated to detect DHX9 or NPM1 binding regions in KIMAT1 sequence. Fragment sizes were confirmed by PCR (bottom panel), and binding of each fragment to DHX9 or NPM1 determined via pull-down of biotinylated-labeled RNA fragments and immunoblotting (IB) (top panels) with the indicated antibodies. BR binding region; nt nucleotide. **h** RNA immunoprecipitation assay. H1299 cells were transfected with FLAG-tagged full-length DHX9 (FL), or DHX9 without the two double-stranded RNA-binding domains (dsRBD Del) or Empty vector (NT), crosslinked and subjected to immunoprecipitation using FLAG-specific antibody. (Top) DHX9-bound KIMAT1 was analyzed by qPCR. (Bottom) Immunoblotting showing immunoprecipitation's efficiency. **i** RNA immunoprecipitation assay. H1299 cells were transfected with FLAG-tagged full-length NPM1 (FL), or NPM1 without the RNA-binding domain (DRBD Del) or Empty vector (NT), crosslinked and subjected to immunoprecipitation with a FLAG-specific antibody. (Top) NPM1-bound KIMAT1 was analyzed by qPCR. (Bottom) Immunoblotting showing immunoprecipitation's efficiency. **j** Immunoblotting showing DHX9 KO and NPM1 KO in two different CRISPR/Cas9 clones. **k** 3D invasion assay and quantification of the spheroid area in the two DHX9 and NPM1 CRISPR/Cas9 clones in **j** compared to control cells (Cas9 only). Scale bar, 500 μm. **l** immunoblotting showing downregulation of DHX9 and NPM1 upon transfection of H1299 cells with two KIMAT1 GpRs. **a, b, e, f, g** Representative results or images of two biological replicates. Mean ± S.D (n = 3). p values were calculated by two-tailed Student's t test.

confirming that NPM1 is part of a complex composed by DHX9 and DDX5. The association of DHX9 and NPM1 with DDX5 was abrogated by treatment with RNase A, indicating that RNA molecules, including KIMAT1, may be important for the binding of DHX9 and NPM1 to the MC (Fig. 6f). However, we still detected a binding between DHX9 and NPM1 in presence of RNASe A. Thus, DHX9 and NPM1 interact in the nucleus independently of KIMAT1 (Fig. 6f). Subsequently, to determine whether DHX9 and NPM1 were able to bind to specific pri-miRNA substrates, we conducted a CLIP assay which confirmed that DHX9 and NPM1 bind to pri-miR-17-5p and -18a and not to pri-miR-200b, -200c, -7, -27a, -27b, -139 and -let-7b (Supplementary Fig. 11d). We also performed an in vivo cellular monitoring

assay of DHX9 and NPM1 function, as previously reported[9]. Two different cell lines were transfected with a luciferase vector carrying a segment of pri-let-27b, -let-7b, -17-5p and -18a between the luciferase gene and the polyadenylation signal (Supplementary Fig. 11e). As consequence of DROSHA-mediated cleavage of the pri-miRNA, the luciferase transcripts lose their polyadenylation tail, resulting in poor stability and decreased translation[9]. In this cellular system the luciferase intensity is inversely correlated to Drosha activity (Supplementary Fig. 11e). This experiment revealed that DHX9 or NPM1 silencing caused an increase in the luciferase activity of cells transfected with pri-miR-17-5p and pri-miR-18a and not of cells transfected with pri-miR-27b or pri-let-7b, while DHX9 OE or NPM1 OE resulted in the opposite effect

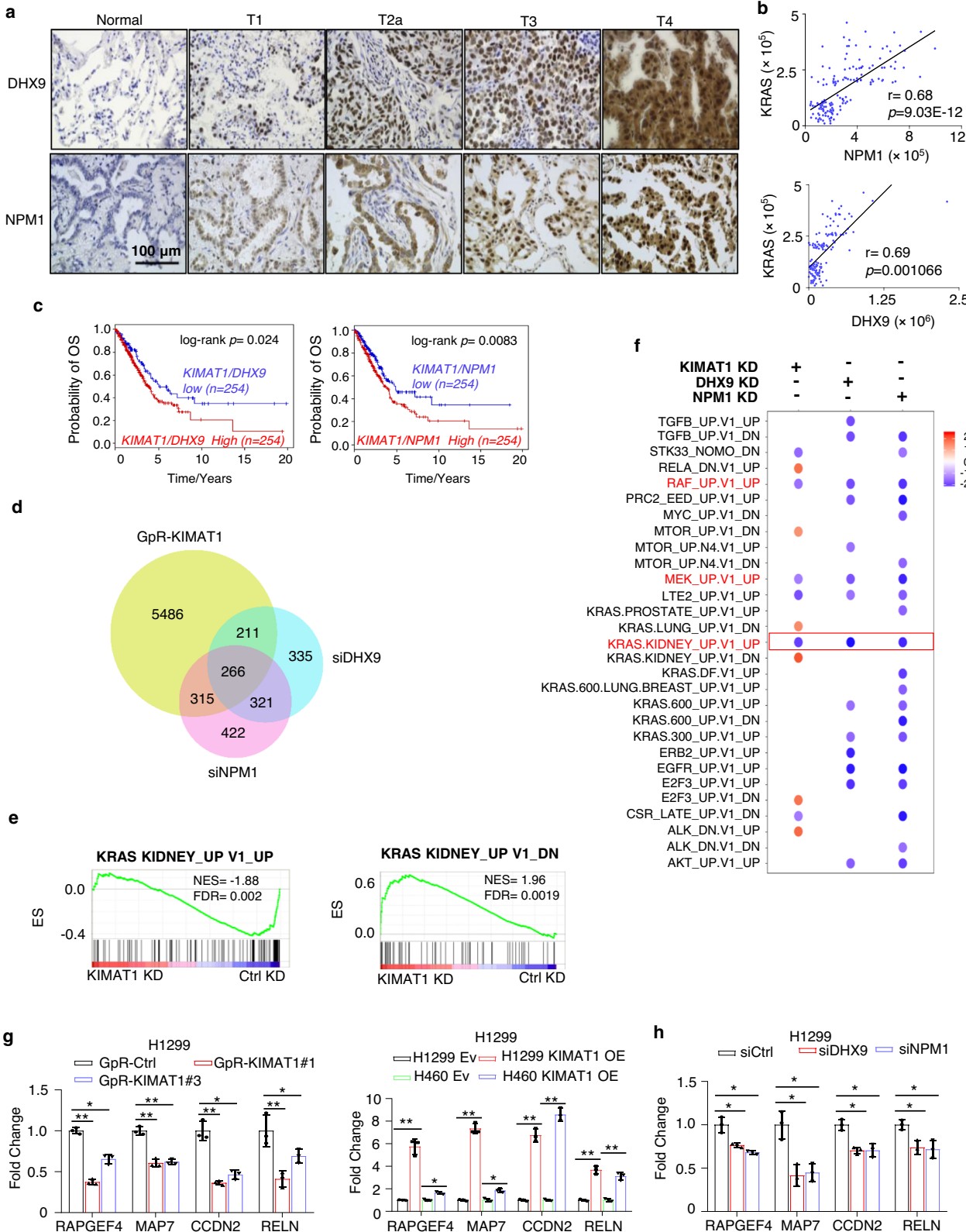

(Supplementary Fig. 11f). As for *KIMAT1*, network analysis evidenced that genes belonging to pathways activated by DHX9 or NPM1 were predicted targets of DHX9- and NPM1-suppressed miRNAs (Fig. 6g and Supplementary Data 4). Altogether, these data suggest that *KIMAT1* enhances the processing of a subset of miRNAs that foster lung cancer progression via DHX9 and NPM1.

**p21 is a component of the microprocessor complex.** Having assessed that *KIMAT1* KD induced downregulation of miR-17-5p, -18a, -375 and 10b-5p via DHX9 and NPM1, we sought to address how *KIMAT1* KD could lead to the induction of miR-200 family members, miR-7, -27a, -27b, -139 and let-7b, considered to have TS function in lung cancer. TP53 has been shown to enhance the post-transcriptional maturation of miRNAs with

**Fig. 5 KIMAT1, DHX9 and NPM1 regulate the KRAS signaling. a, b** DHX9 and NPM1 are overexpressed in LUAD lesions ($n = 75$) at different stage (T1–T4) compared to matched normal lung samples and directly correlate with KRAS expression in the same samples. $p$ values were calculated by two-tailed Student's $t$ test. **c** Kaplan–Meier survival curves comparing survival of patients with high KIMAT1/DHX9/NPM1 expression (red line) to patients with low KIMAT1/DHX9/NPM1 expression (blue line) from the TCGA LUAD dataset. The log rank $p$ values were obtained from a two-tailed Chi-Square test. **d** Venn diagram depiction of the overlap of significant differentially expressed genes between KIMAT1 and its interacting proteins and between DHX9 and NPM1. **e** Enrichment plots of gene sets positively or negatively regulated by KRAS upon KIMAT1 KD. **f** Gene signature analysis of KIMAT1-, DHX9- and NPM1-related pathways in KIMAT1 KD, DHX9 KD and NPM1 KD cells. Highlighted in red are the common signatures between KIMAT1, DHX9 and NPM1. $p < 0.05$ by two sided Kolmogrorov–Smirnov (K-S) test. The resulting $p$ values were corrected for multiple hypothesis testing. **g, h** Bar plot depiction of the expression levels of randomly selected genes belonging to the KRAS signaling commonly modulated by KIMAT1, DHX9 and NPM1 by qRT-PCR. **g** (Left) $p$ values from left to right= 2.88E−05, 0.001022, 0.000675, 0.000367, 0.0008, 0.002051, 0.0001, 0.0168; **g** (Right) $p$ values from left to right= 0.000316, 0.001236, 2.23E−05, 0.002561, 6.74E−05, 2.27E−05, 0.000226, 0.000531. **h** $p$ values from left to right *0.0119, 0.003881, 0.00756, 0.007485, 0.0022, 0.0064, 0.0092, 0.014. Error bars represent mean ± S.D ($n = 3$). $p$ values were calculated by two-tailed Student's $t$ test.

growth suppressive functions[8,40]. However, RNA-seq analysis (Supplementary Fig. 9a) was carried out in H1299 cells, which are TP53-null. Thus, we searched for other potential tumor suppressor genes derepressed after KIMAT1 KD and confirmed a significant MYC-dependent upregulation of CDKN1A (p21) (Fig. 7a and Supplementary Fig. 12a–c)[41]. Interestingly, network analysis evidenced that MYC is a target of several KIMAT1-repressed miRNAs, pointing to a post-transcriptional regulation of MYC by KIMAT1 (Supplementary Fig. 12d). p21 attracted our attention because it regulates multiple tumor suppressor pathways in a p53-independent fashion[42] and its expression is reduced in late compared to early stage lung cancer, suggesting that p21 may be downregulated in more aggressive tumors[43]. Therefore, we investigated whether p21 could be responsible for the induction of KIMAT1-repressed miRNAs. p21 KD induced marked upregulation of oncogenic-like miRNAs and down-regulation of miRNAs with growth suppressive function, whereas p21 enforced expression caused the opposite effect (Fig. 7b). Importantly, the levels of the precursor miRNAs changed while the primary remained unaffected, further substantiating that p21 may be involved in miRNA biogenesis (Fig. 7b). Because KIMAT1 KD induces MYC downregulation, we were surprised not to see an effect on pri-miR-17-5p and pri-miR-18a, members of the well-known MYC-induced miR-17-92 cluster[44] upon KIMAT1 silencing. Strikingly, pri-miR-17-5p and pri-miR-18a remained unaltered in cells transfected with a MYC siRNA, suggesting that in our system MYC does not transcriptionally regulate members of this cluster (Supplementary Fig. 12e). We then tested whether p21 could directly interact with members of the MC. Reverse co-immunoprecipitation revealed an interaction between endogenous Drosha and p21 in H1299 (p53 null) and A549 (p53WT), confirming that p21 is a Drosha interactor (Fig. 7c).

CLIP assay revealed that p21 binds to pri-miR-200b, -200c, -27b and let-7b and not to pri-miR-17-5p, 18a, -375 and -10b (Supplementary Fig. 12f). To further verify a role for p21 in miRNA processing, we performed an in vivo cellular monitoring assay. H1299 and H460 cell lines were transfected with a luciferase vector carrying a segment of pri-27b, pri-let-7b, pri-miR-17 and pri-miR-18a between the luciferase gene and polyadenylation signal[9]. p21 silencing caused an increase in the luciferase activity in cells transfected with pri-27b and let-7b but not in cells transfected with miR-17 and miR-18a, while p21 OE resulted in the opposite effect (Supplementary Fig. 12g). Thus, p21 may enhance the processing of a subset of miRNAs with tumor suppressor function. In a rescue experiment, p21 OE reduced the binding between primary oncogenic miRNAs and DHX9 and between primary oncogenic miRNAs and NPM1 (Fig. 7d). Moreover, overexpression of p21 in A549 or H1299 cells hampered the binding between DDX5 and DHX9 and between DDX5 and NPM1 (Fig. 7e). These results pointed to a possible antagonistic effect between components of the MC for the

binding to pri-miRNAs and for triggering their cleavage by Drosha. Investigating this antagonistic effect further, we noticed that p21 OE decreased DDX5 and NPM1 protein but not mRNA levels (Fig. 7f and Supplementary Fig. 12h). Thus, p21 OE impairs the binding between DHX9 and DDX5 and between NPM1 and DDX5 by reducing DDX5 and NPM1 expression levels post-transcriptionally. Conversely, DHX9 KO and NPM1 KO decreased MYC expression with consequent p21 upregulation, suggesting that in absence of DHX9 and NPM1, p21 binds to Drosha to promote the processing of TS miRNAs (Supplementary Fig. 12i). Importantly, p21 is downregulated in LUAD and LUSC lesions (Supplementary Fig. 13a). p21 silencing increased the capacity of the cells to proliferate, whilst p21 OE induced significant cell death (Supplementary Fig. 13b and Fig. 7g), revealing that the substantial apoptosis observed upon KIMAT1 KD could mainly be mediated by p21. Notably, pro-apoptotic genes induced by KIMAT1 KD (Supplementary Fig. 13c) were also induced by p21 OE in two NSCLC cell lines (Supplementary Fig. 13d). Treatment with the transcription inhibitor Actinomycin D did not affect p21-induced cell death, further substantiating that p21-mediated effect on apoptosis is post-transcriptional (Supplementary Fig. 13e). Altogether, these findings indicate that p21 promotes the processing of TS miRNAs and its OE induces cell death by post-transcriptional modulation of gene expression.

**KIMAT1 silencing halts tumor growth in a PDX mouse model while its overexpression promotes lung cancer progression.** Having assessed a role for KIMAT1 in the regulation of KRAS signaling, we evaluated the biological effects of KIMAT1 modulation in lung tumors. GpRs have recently been shown to be effective in targeting RNA in vivo[24,45]. To test the therapeutic potential of KIMAT1 targeting in vivo, we employed a patient-derived xenograft (PDX) lung squamous cell carcinoma model harboring 6 copies of KRAS[WT] from a patient with lymph node metastasis (Supplementary Fig. 14a). Intravenous injections of KIMAT1 GpRs alone significantly inhibited tumor growth, reduced KRAS and Ki67 protein levels and induced cell death in vivo, as revealed by the increase in cleaved caspase 3 (Fig. 8a, b and Supplementary Fig. 14b). Importantly, tumors treated with KIMAT1 GpRs showed remarkable downregulation of KIMAT1, DHX9 and NPM1 and increased expression of p21 (Supplementary Fig. 14c). Furthermore, to assess the metastatic potential of cells overexpressing KIMAT1, we injected H1299 cells with stable expression of KIMAT1 in the tail vein of NSG mice, as previously described[46,47]. 17 weeks later metastases and micro-metastases were observed in the lungs and liver of the majority of mice injected with KIMAT1 stable cells compared to controls (Supplementary Fig. 15a). We also performed orthotopic injections of H1299/luc2+/KIMAT1 and H460/luc2+/KIMAT1 stable cell lines in the lungs of NSG mice and evaluated the capacity of the cells to metastasise over time[48,49]. KIMAT1 OE promoted

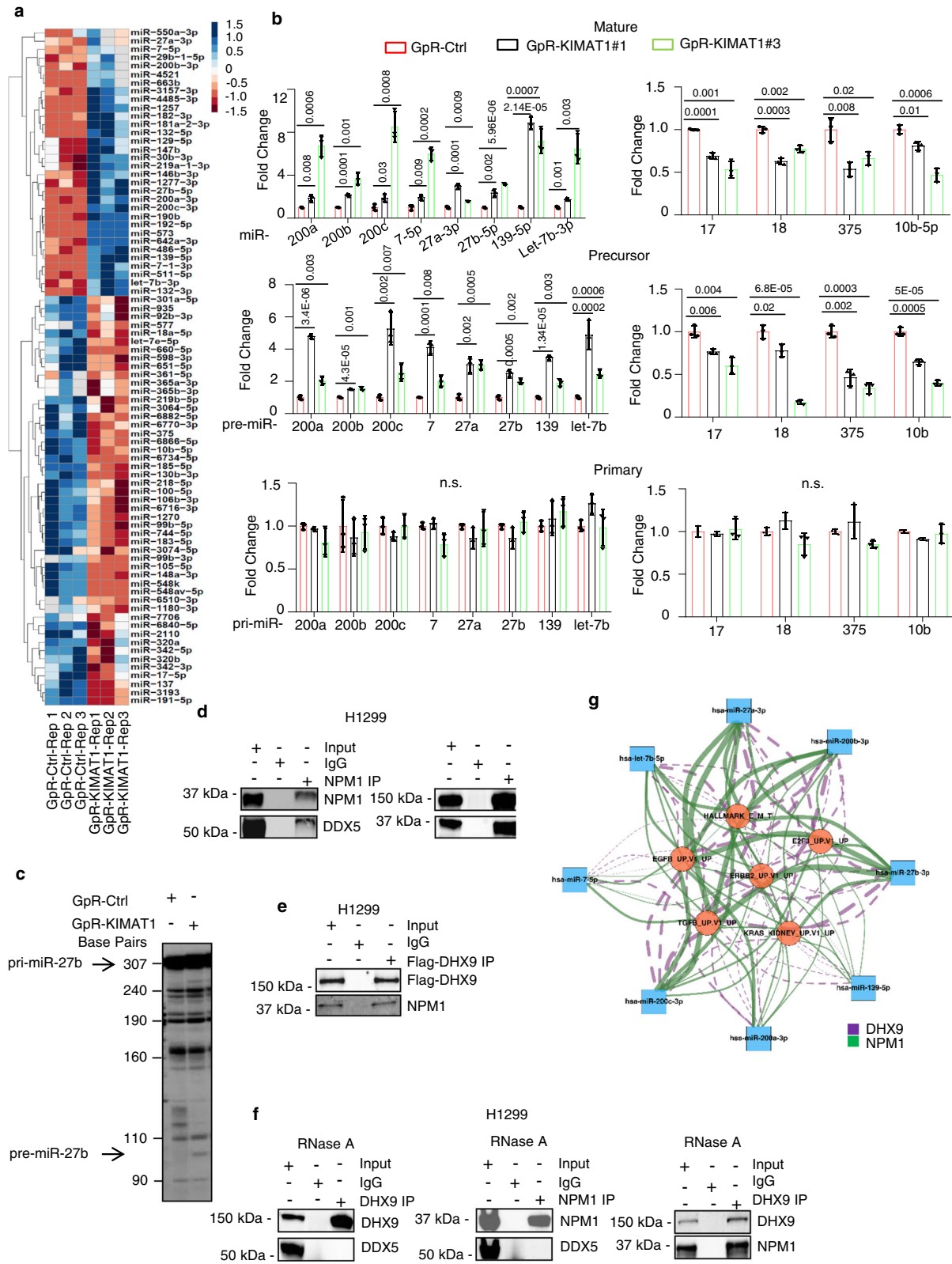

tumor initiation and gave rise to malignant ascites and metastases in liver and kidneys (Fig. 8c, d and Supplementary Fig. 15b–d). Reciprocally, DHX9 KO and NPM1 KO reduced KRAS expression and the number of distant metastases in vivo (Fig. 8e and Supplementary Fig. 16a–c). In a rescue in vivo experiment, orthotopic injection of cancer cells simultaneously overexpressing *KIMAT1* and harboring DHX9 or NPM1 deletion *(DHX9 KO or*

NPM1 KO) halted *KIMAT1*-mediated metastatic effects, suggesting that DHX9 and NPM1 are important mediators of *KIMAT1* function (Supplementary Fig. 17a–d).

## Discussion
In this study, we have shown that KRAS$^{WT}$ amplification is a marker of poor prognosis in both LUAD and LUSC. KRAS$^{WT}$ OE

**Fig. 6 *KIMAT1* regulates miRNA processing. a** Hierarchical clustering analysis of dysregulated miRNAs upon *KIMAT1* KD. A padj ≤0.05 defined genes significantly differentially expressed. Log$_2$fold change (FC) > 0.48, < −0.27. Dysregulated miRNAs were obtained using the DESeq2 method, which utilizes Negative Binomial GLM fitting and two-sided Wald test to compare two groups for hypothesis testing. The *p* values from the Wald test were corrected for multiple testing using the Benjamini and Hochberg method. **b** Expression levels of mature, precursors and primary miRNAs with tumor suppressor (left) or oncogenic function (right) in *KIMAT1* KD cells by qRT-PCR. Primary and precursor miRNAs were normalized to *β-actin*, mature miRNAs were normalized to *RNU48*. Error bars represent mean ± S.D (*n* = 3). \*\**p* value < 0.001, \**p* value < 0.05 by two-tailed Student's *t* test. **c** *KIMAT1* silencing induces Drosha-mediated *pri-miR-27b* in vitro processing. *n* = 2. Pre-miR-27b band size = 97 nt. **d** Reverse IP demonstrating the interaction between DDX5 and NPM1 and between NPM1 and DHX9 endogenous levels. Representative images of 2 biological replicates. **e** DHX9/NPM1 interaction following transfection of H1299 cells with a DHX9-Flag plasmid and subsequent immunoprecipitation using a Flag-specific antibody. Representative images of 2 biological replicates. **f** Treatment with RNase A abrogates DHX9/DDX5 and DHX9/NPM1 interaction but not the interaction between DHX9 and NPM1. Representative images of 2 biological replicates. **g** Directed Network Diagram of Enriched Pathways regulated by DHX9 and NPM1 with Predicted miRNA Targets derived from each KD experiment (DEGs with FDR < 0.05 which are also predicted to be miRNA targets by Targetscan [release 7.2 – default predictions]). The statistical method to obtain the DEGs has been described in the legend of Fig. 6a. The diagram displays miRNAs (blue squares) and pathways enriched (orange circles) (FDR < 0.05; calculated using GSEA with the DEGs from DHX9/NPM1 KD) and the connecting edges (purple-dashed: DHX9; green-solid: NPM1); edge-widths represent the number of DEGs contributing to the connection within each gene set. The network graph was produced using the igraph package (version 1.2.4.2).

is able to activate in vitro the same oncogenic pathways as mutant KRAS, fostering cell migration and invasion and therefore, lung cancer progression. LncRNAs are important regulators of gene expression involved in several processes, spanning from transcriptional to post-transcriptional regulation, mRNA splicing and decay[3,4]. Here, we leveraged the pleiotropic functions of lncRNAs to identify KRAS-regulated pathways that could be exploited for therapeutic purpose. We characterized the top induced lncRNA upon KRAS OE, *KIMAT1*, and demonstrated that it correlates with KRAS expression in vitro and in vivo. Notably, high levels of *KIMAT1* and KRAS were observed in cells derived from metastatic sites and in late stage lung adenocarcinomas. Consistent with the hypothesis that KRAS amplification propels tumor propagation, our experiments revealed that MYC, a well-known KRAS-activated transcription factor fundamental for KRAS-induced tumorigenesis[18], binds to *KIMAT1* promoter determining its activation. Inspection of the *KIMAT1* locus evidenced that it is essentially composed of TEs, and MYC-dependent transcription of *KIMAT1* is driven by a LTR originated from evolutionary ERV1 integration events in the human genome. *KIMAT1* silencing reduced 3D cell invasion and induced substantial programmed cell death in cancer and not in normal cells, suggesting that it is essential for cancer survival. Although *KIMAT1* expression is lower in cells with a mutational KRAS status compared to those with amplified KRAS, the biological effects upon *KIMAT1* manipulation in these cells are similar to those observed in cells with KRAS amplification. Furthermore, by RAP-MS and CLIP assay we discovered that *KIMAT1* binds to and stabilizes DHX9 and NPM1, which are present in both the nucleus and cytoplasm[22,50], in accordance with *KIMAT1* localization. DHX9 is overexpressed in many cancers but it is also fundamental in maintaining normal cellular homeostasis[22]. NPM1 is regarded as an effective therapeutic target for the treatment of both solid and hematologic malignancies. Nevertheless, none of the molecules targeting NPM1 discovered in the last few years have shown chemical features suitable for their development as drugs[51,52]. *KIMAT1* is detectable in cancer and has very little or no expression in normal tissues. Thus, targeting *KIMAT1* would allow DHX9 and NPM1 silencing only in tumors, without toxicity for the normal counterpart. MiRNAs are important regulators of gene expression in physiological and pathological processes, including cancer[7,53]. They can function as both tumor suppressors and oncogenes depending on the cellular context[54,55]. Our findings raise the possibility that KRAS is a upstream regulator of the MC complex and regulates its own signaling at least in part post-transcriptionally by governing miRNA processing. *KIMAT1*, through DHX9 and

NPM1 stabilization and MYC-dependent suppression of p21, promotes the processing of a subset of oncogenic-like miRNAs while simultaneously halting the biogenesis of a subset of miRNAs with tumor suppressor function, maintaining a positive feedback loop that potentiates the KRAS signaling (Fig. 8h). To our knowledge, this antagonistic effect on miRNA processing by members of the MC to promote or suppress tumorigenesis has hitherto never been reported. Further investigation would be fundamental in defining the effective number of pri-miRNAs that bind to DHX9/NPM1 or p21 and how the recognition occurs. Notably, previous studies reported that components of the MC complex are maintained together by RNA molecules[8]; however, none of these RNAs have so far been identified. Here, we demonstrated that *KIMAT1* is fundamental for the binding of DHX9 and NPM1 to DDX5 and is therefore essential for miRNA processing. In summary, we reported a so far unidentified network downstream of KRAS that could be exploited for therapeutic intervention to halt the adaptability of KRAS-driven tumors. Given the recent promising results with RNA therapeutics[56], the important role of *KIMAT1* in lung tumorigenesis and the robust effect in suppressing the growth of a patient-derived tumor harboring KRAS$^{WT}$ amplification, we put forward the idea that its targeting may be effective in the treatment of a subset of lung cancer patients for whom, at the moment, there are no effective treatment options.

## Methods

**Murine models**. Animal experimental procedures were approved by Cancer Research UK Manchester Institute's Animal Welfare and Ethical Review body in accordance with the Animals Scientific Procedures Act 1986 and according to the ARRIVE guidelines and the Committee of the National Cancer Research Institute guidelines. All the in vivo studies, except the PDX mouse model, which was carried out by Xentech (Evry, France), were conducted under the project license number P72E31537 (M.G.). The PDX used was not generated for the purpose of this study. The authorization (PDX mouse model) to use animals in the Center for Exporation and Experimental Functional Research (CERFE, Evry, France) facility was obtained by The "Direction of the Veterinarian Services, Ministry of Agriculture and Food, France" (agreement No. D-91-228-107). All experiments are performed in accordance with French legislation concerning the protection of laboratory animals and in accordance with a currently valid license for experiments on vertebrate animals, issued by the French Ministry of Higher Education, Research and Innovation. Pascal Leuraud is in charge of the implementation and compliance to the Projects Agreements: «Evaluation de l'activité antitumorale de candidats médicaments en monothérapie sur des modèles de xénogreffes de tumeurs dérivées de patients»; APAFIS#14073-2018021311396396 v2 (valid for 2 years from September 09th 2018). Mice were observed for signs of illness or distress during the course of the experiments and body weight was measured twice a week. Animals were euthanized after the appearance of predefined criteria like rapid weight loss (>20%) or weight gain (>20% due to ascites) and labored respiration. After euthanasia mice were analyzed for the presence of peritoneal tumors. Lungs, liver and kidneys were excised, weighed, photographed and bisected. Part of the organs was fixed in

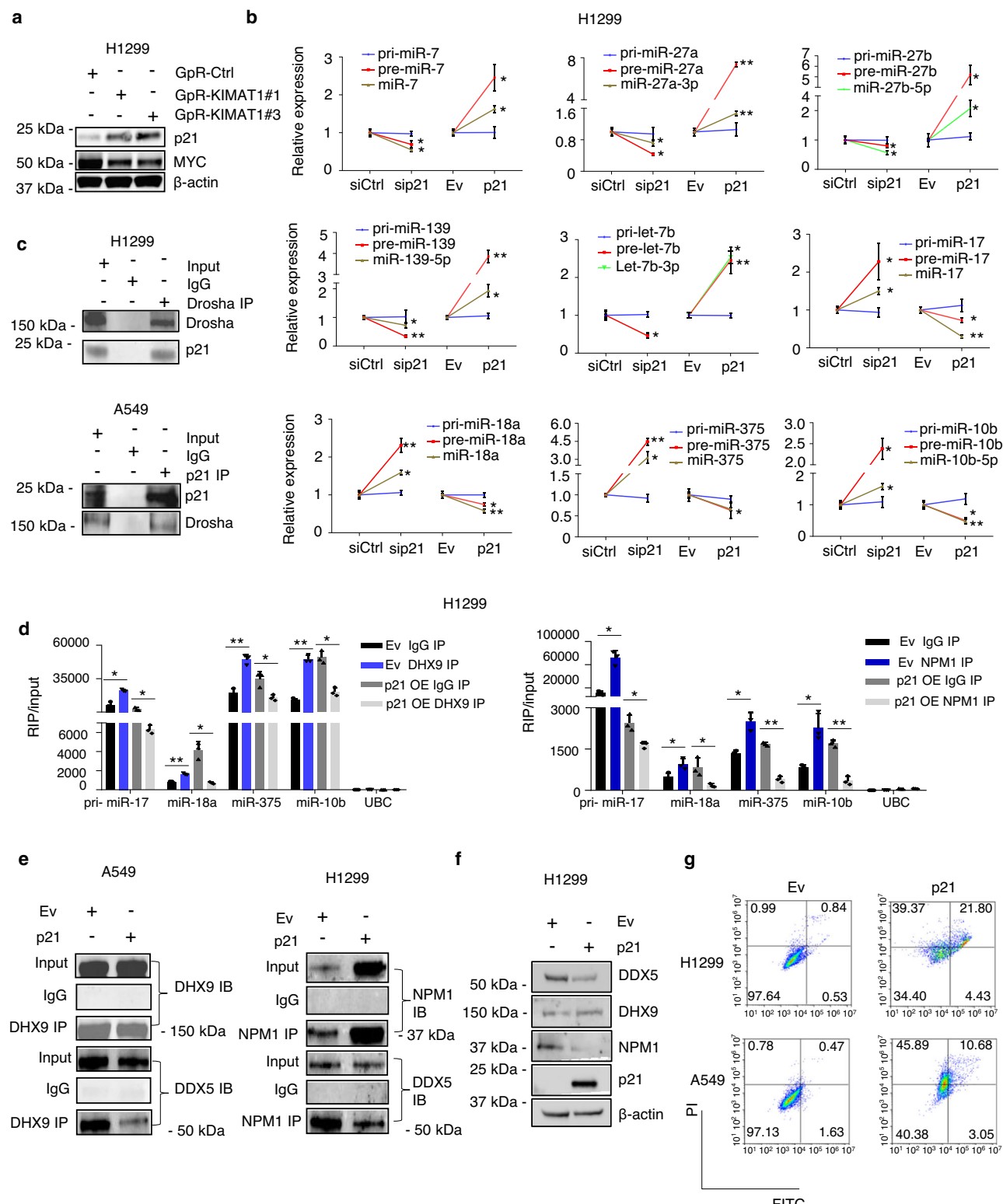

formalin and subjected to immunohistochemistry (IHC) for further analysis. IHC quantification was determined by QuPath (v0.0.0-m2). The remaining organ was snap-frozen in liquid nitrogen and stored at −80 ºC. RNA was isolated from tissues with TRIzol solution and qPCR was performed to analyze genes and miRNA expression.

**Murine subcutaneous in vivo model**. $5 \times 10^6$ H1299 and H460 cells stably expressing a control vector (Ev) or *KIMAT1* were subcutaneously injected into the right posterior dorsal flank of 4–6 weeks old female NOD/SCID Gamma (NSG) mice ($n = 8$ per group) (Charles River). All animals were maintained in a

pathogen-free environment with free access to food and water. Tumor size was assessed twice per week using a digital caliper by measuring the length ($l$) and the width ($w$) and calculated based on the formula $V = lw^2/2$. Mice were euthanized and sacrificed when tumor size reached the endpoint of 1500 mm$^3$ or mice presented signs of illness.

**Patient-Derived Xenograft (PDX) mouse model**. Female NSG mice, weighting 20–30 g, 4–6 weeks old were maintained in specific pathogen-free animal housing. The patient-derived xenograft (IC11LC13), from a patient with a lung squamous cell carcinoma harboring 6 copies of KRAS$^{WT}$ and loss of function mutation in p53

**Fig. 7 p21 is a component of the MC and antagonizes DHX9 and NPM1 effect on miRNA biogenesis. a** *KIMAT1* KD induces MYC downregulation and p21 upregulation. Representative images of 2 biological replicates. **b** Expression levels of primary, precursors and mature forms of the indicated miRNAs were analyzed by qPCR after silencing or overexpression of p21 in H1299 cells. Primary and precursor miRNAs were normalized to *β-actin* and mature miRNAs were normalized to *RNU48*. sip21 vs siCtrl pre-miRNA and mature miRNA p value= 0.013 (pre-miR-7), 0.003 (miR-7-5p), 0.001 (pre-miR-27a), 0.018 (miR-27a-3p), 0.0496 (pre-miR-27b), 0.008 (miR-27b-5p), 0.00015 (pre-miR-139), 0.008 (miR-139-5p), 0.004 (pre-let-7b), 0.0011 (let-7b-3p), 0.011 (pre-miR-17), 0.0024 (miR-17), 0.0004 (pre-miR-18a), 0.001 (miR-18a), 3.448E-05 (pre-miR-375), 0.0015 (miR-375), 0.001 (pre-miR-10b), 0.0014 (miR-10b-5p); p21 OE vs Ev pre-miRNA and mature miRNA *p* value = 0.002 (pre-miR-7), 0.001 (miR-7-5p), 1.06E-06 (pre-miR-27a), 0.0002 (miR-27a-3p), 0.0014 (pre-miR-27b), 0.007 (miR-27b-5p), 7.9E-05 (pre-miR-139), 0.002 (miR-139-5p), 0.002 (pre-let-7b), 0.0001 (let-7b-3p), 0.0019 (pre-miR-17), 0.0001 (miR-17), 0.0009 (pre-miR-18a), 0.003 (miR-18a), 0.03 (pre-miR-375), 0.002 (miR-375), 0.005 (pre-miR-10b), 0.0007 (miR-10b-5p). **c** p21 interacts with Drosha in p53 null and p53 wild-type cells. Representative images of 2 biological replicates. **d** CLIP analysis of the association between the indicated primary miRNAs and DHX9 or NPM1 upon p21 OE. Cells were transfected with p21 and immunoprecipitated with anti-DHX9 or anti-NPM1 antibody. *Pri-miR-17-5p/miR-18a/miR-375/miR-10b* expression was analyzed by qRT-PCR. *UBC* was used as a negative control. Error bars represent mean ± S.D (n = 3), **p value < 0.001, *p value < 0.05 by a two-tailed Student's t test. (Left) From left to right p values = 0.0013436, 0.001265, 0.0004582, 0.002409, 0.000873, 0.014035, 7.022E−05, 0.001565; (Right) From left to right p values = 0.0013656, 0.013152, 0.0321765, 0.029651, 0.0043712, 6.92E−05, 0.0088363, 0.000348. **e** p21 OE abrogates the interaction between DHX9 and DDX5 and between NPM1 and DDX5. Representative images of 2 biological replicates. **f** p21 enforced expression reduces DDX5 and NPM1 protein level. Representative images of 2 biological replicates. **g** p21 OE in p53 null or wild-type cells induces prominent cell death.

was transplanted subcutaneously onto mice. When tumors reached around 150 mm³, the tumor-bearing mice were randomly divided into equivalent groups of 7 animals. Mice were treated with 20 mg/kg of GpR *KIMAT1* (n = 7) or vehicle (n = 7), i.v. three times per week (the first two weeks) and then twice a week (the last two weeks) for a total of 10 injections. All mice were weighted twice weekly. Tumor growth was monitored by measuring tumor diameters with a caliper. Tumor volume (V) was calculated as follows: V = a² × b/2, where a is the width (large diameter) and b the length (small diameter) of the tumor in millimeters. Two tailed Student's *t* test was used to measure statistical significance.

**Mouse model of metastasis**. $5 \times 10^5$ H1299/*KIMAT1* or H1299/Empty vector (H1299-Ev) cells in 0.2 ml PBS were injected in the tail vein of NSG mice (n = 7). Seventeen weeks later, mice were sacrificed and lungs and livers were collected for further analysis.

**Orthotopic mouse model**. NSG mice were anesthetized with isofluorane and placed in the right lateral decubitus position. $2.5 \times 10^6$ H1299 or H460 cells stably expressing *KIMAT1* or a control vector (Ev) and the luciferase gene (*luc2+*), or H1299 DHX9 KO, NPM1 KO cells using the CRISPR/Cas9 editing system were injected with a 0.5 ml insulin syringe percutaneously into the left lateral thorax, at the lateral dorsal axillary line of 5–7 weeks old NSG mice (8 mice per group). After injection mice were transferred to a clean recovery cage on top of a heated mat and observed until they fully recovered. Primary tumors and/or metastases were examined over time using luminescence imaging (IVIS). Mice were sacrificed 5 weeks after the injection of the cells. Lungs, livers and kidneys were collected after autopsy for histological analysis. Bioluminescent signal was quantified using IVIS Spectrum in vivo Imaging System. Two tailed Student's *t* test was used to measure statistical significance.

**Cell lines**. Lung adenocarcinoma cell lines H1299, H460, A549, H1975, CALU1 and CALU6, lung squamous cell carcinoma cell line H520, lung fibroblasts HEL299, lung bronchial epithelial cell line HBEC3-KT, normal human bronchial epithelium BEAS2B cells, kidney embryonic cells HEK293 were purchased from American Type Culture Collection (ATCC) and cultured as suggested by ATCC's guidelines. CORL-23 cells were purchased from Sigma-Aldrich. Type II pneumocytes cells were a kind gift of Prof. Julian Downward (The Institute of Cancer Research, London).

**Human tissue samples**. LUAD (HLugA150CS02) and LUSC (HLugSqu150SC01) microarrays (TMA) were purchased from US biomax. Single-molecule RNA FISH (smFISH) and IHC were used to examine the expression of *KIMAT1*, KRAS, DHX9 and NPM1 in 75 tumor and matched normal lung samples (LUAD normal n = 75, T1 = 37, T2a = 14, T2b = 4, T3 = 16, T4 = 4; LUSC normal n = 75, T1 = 23, T2a = 13, T2b = 10, T3 = 24, T4 = 5). Images were acquired with gSTED microscope and spots were counted using the online JAVA software of StarSearch[57].

**Cell fractionation**. Total nuclear and cytoplasmic extracts were obtained from cells cultured in 60-cm dishes using the Cytoplasmic and Nuclear RNA Purification Kit (Norgen), according to the manufacturer's instructions.

**CPAT (Coding Potential Assessment)**. Coding potential for *KIMAT1* was determined using the CPAT tool[16]. *KRAS* and *MALAT-1* were used as controls.

**RACE**. *KIMAT1* 5' and 3' ends were identified by Rapid amplification of cDNA ends (RACE) using the Roche 5'/3' Kit according to manufacturer's instructions. Briefly, RNA was extracted from H1299 cells and cDNA was synthesized using a GSP1 primer or oligo (dT)-Anchor primer. RACE PCRs were amplified using High Fidelity polymerase and separated on a 1% agarose gel. Gel purified products were cloned in a pCDH vector and sequenced to identify the 5' and 3' ends of the transcript. Primers are listed in Supplementary Table 3.

**Gene editing via CRISPR/Cas9 system**. Single guide RNA (sgRNA) sequences targeting different segments of DHX9 genes were designed using the CRISPR design tool (http://tools.genome-engineering.org)[58]. sgRNAs were inserted in a GFP plasmid containing the Cas9 and the sgRNA scaffold (pSpCas9(BB)2A-GFP) (addgene) digested with BbsI (ThermoFisher Scientific). The inserted sequences were verified by sequencing. H1299 cells were transfected with the CRISPR/Cas9 constructs using Lipofectamine 2000 regent for 48 h and sorted based on GFP expression using Flow Cytometry (Novocyte NovoExpress Software, version 1.3.0). Primers are listed in Supplementary Table 3.

**LncRNA cloning and lentiviral transduction**. *KIMAT1* full-length or deletion constructs were PCR amplified and inserted into a lentiviral vector (pCDH-GFP, System Biosciences). The inserted sequences were validated by sequencing. Cells were sorted based on GFP expression using Flow Cytometry (Novocyte NovoExpress Software, version 1.3.0). Primers are listed in Supplementary Table 3.

**Dual-Luciferase reporter assay**. Two LTRs upstream of *KIMAT1* TSS containing two and one MYC BS respectively, were PCR amplified and inserted into a promoterless pGL3 basic vector (Promega). 200 ng of pGL3 basic vector, 20 ng of Renilla plasmid (Promega), and 50 nM of MYC siRNA (Applied Biosystems) were co-transfected in H1299 or CALU6 cells for 48 h. Dual-Luciferase Assay (Promega) was used to examine luciferase activity. Primers are listed in Supplementary Table 3. Deletion of MYC-binding sites was obtained using the QuickChange Mutagenesis Kit (Stratagene).

**In vivo monitoring of pri-miRNA processing**. The firefly luciferase pmirGLO Plasmid (Promega) containing pri-miRNA sequences (primiR-27b, let-7b, miR-17, miR-18a) at the 3' untranslated region, a Renilla plasmid and DHX9/ NPM1 siRNAs or DHX9/NPM1 expression vectors were co-transfected into H1299 or H460 cells for 48 h. Dual-Luciferase Assay (Promega) was performed to examine the reporter activity. Primers are listed in Supplementary Table 3.

**In vitro pri-miRNA processing**. In vitro pri-miRNA processing assay was performed as previously described[59]. Pri-miR-27b containing the stem-loop sequence plus 150 flanking bp was PCR amplified with a T7 containing forward primer and purified PCR product in vitro transcribed in the presence of α³²P CTP (PerkinElmer). $10^5$ cpm isotope-labeled pri-miRNA was incubated at 37 °C for 90 min with a reaction containing 30 μl of Drosha IP complex, 3 μl of 10× reaction buffer (64 mM MgCl₂), 0.75 μl of RNase inhibitor (Roche) in RNase free water. RNA was extracted with phenol/chloroform and precipitated with ethanol overnight. RNA was loaded on a 6% denaturing polyacrylamide gel and then exposed with XAR-5 autoradiography film (KODAK) overnight at −80 °C with an intensifying screen. Pri-miRNA cloning primers are listed in Supplementary Table 3.

**Protein extraction and immunoblotting**. Total protein lysates were homogenized in 1× RIPA buffer (Sigma-Aldrich) plus protease inhibitors (Roche) and centrifuged with

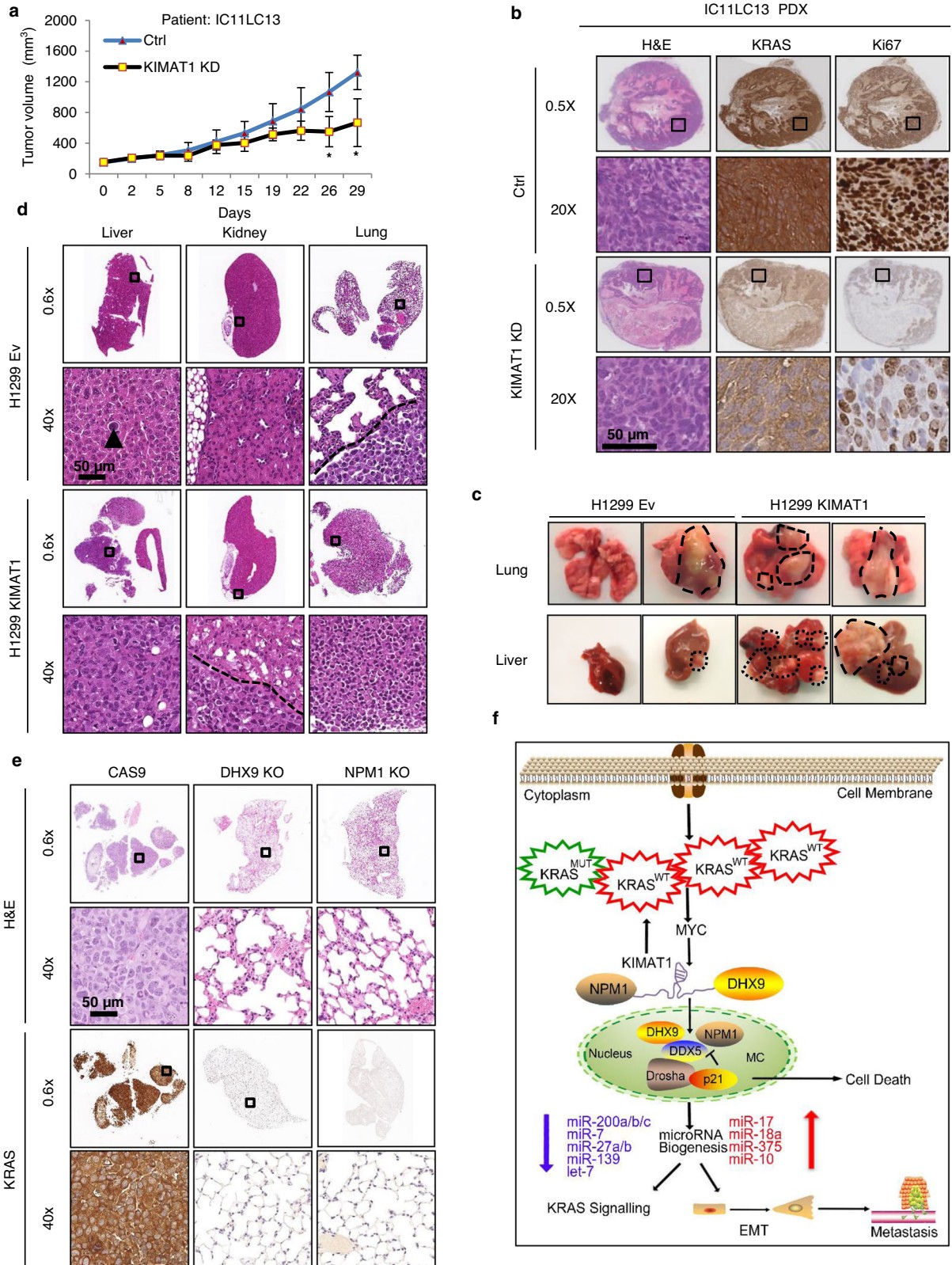

13,000 × g for 20 min at 4 °C. Protein concentration was measured by SpectraMax M5 (SoftMax Pro6) using the Pierce BCA Assay Kit (ThermoFisher Scientific). Supernatant was used for immunoblotting with the indicated antibodies. Signal was detected using Western Bright ECL-Spray substrate and ChemiDoc instrument (Bio-Rad).

**RNA extraction and quantitative real-time PCR (qRT-PCR).** Total RNA was isolated from cells and tissues using TRIzol according to the manufacturer's instructions. For miRNA detection, 50 ng of RNA was reverse-transcribed using

Taqman™ miRNA Reverse Transcription Kit and miRNA specific probes (Applied Biosystems™). For gene detection, cDNA was synthesized from 1 µg of total RNA using Verso cDNA Synthesis Kit (ThermoFisher Scientific), and real-time PCR was performed using SYBR Green PCR master mix (Applied Biosystems™). Relative miRNA or gene expression was calculated using ΔCT normalized to *RNU48* or *β-actin*, respectively. Primer sequences are listed in Supplementary Table 4. RNA integrity was verified on an Agilent Bioanalyzer 2100 (Agilent Technologies, Palo Alto, CA).

**Fig. 8 KIMAT1 silencing halts tumor growth in vivo while its OE promotes tumorigenesis and distant metastases. a** Tumor volume of IC11LC13 PDX mice treated by intravenous injections with vehicle (Ctrl) or GpR-KIMAT1 (Error bars represent mean ± S.D, p value by a two-tailed Student's t test.). Ctrl n = 6, KIMAT1 KD n = 3. *0.01194 and 0.002246. **b** Representative IHC images of KRAS and Ki67 levels in lungs of mice treated with KIMAT1 GpRs compared to controls. **c** Representative lungs and livers from NSG mice percutaneously injected into the left lateral thorax with H1299luc + cells stably expressing KIMAT1 or an empty vector. Tumors are evidenced by dashed lines. n = 7. **d** Representative examples of H&E staining of livers, kidneys and lungs from mice injected with KIMAT1 stable cells as in **c**. A single isolated tumor cell is present (arrow head) in the liver of a H1299/Ev mouse. No neoplastic cell is present in a H1299/Ev mouse kidney, whereas a small neoplastic lesion is observed within its lung (under the dashed line). Mice injected with H1299/KIMAT1 cells presented diffuse metastatic infiltration in the livers, peri-renal fat tissue (under the dashed line) and the lungs (mostly entirely substituted by neoplastic tissue). Original magnifications 0.6× and 40×; scale bar, 50 μm. n = 7. **e** Representative H&E and KRAS staining of lung sections from NSG mice percutaneously injected with DHX9 KO or NPM1 KO cells. Original magnifications 0.6× and 40×; scale bar, 50 μm. n = 8. **f** Schematics depicting KIMAT1's mechanism of action. KRAS amplification activates KIMAT1 via MYC-mediated transcription. KIMAT1 binds to and stabilizes DHX9 and NPM1, which promote the processing of oncogenic miRNAs sustaining the KRAS signalling in a positive feedback loop. p21 antagonizes the binding between DDX5 and DHX9 and between DDX5 and NPM1 fostering the processing of tumor suppressor miRNAs which halt KRAS signalling and EMT.

**Chromatin immunoprecipitation (ChIP).** ChIP assay was performed as previously described[60]. Online databases Encode and UCSC genome browser were used to visualize the H3K4me3, H3K27ac and MYC ChIP-seq signals in KIMAT1 promoter. Primers and antibodies are listed in Supplementary Table 4 and Supplementary Table 5.

**siRNA, LNA^TM GapmeR and plasmid transfection.** Commercially available ON-TARGET plus smart pools siRNA for EGFR, MYC, CDKN1A and NPM1 were purchased from Dharmacon and their sequences are listed in Supplementary Table 1. siRNA for DHX9, KRAS were purchased from Thermofisher Scientific. Three different LNA^TM GpRs for KIMAT1 were designed and synthesized from Qiagen. GapmeRs (50 nM) and siRNAs (50 nM) were transfected using Hiperfect reagent (Qiagen) for 48 h according to the manufacturer's instructions. Plasmids were transiently transfected into the cells with Lipofectamine 2000 reagent (ThermoFisher Scientific). H1299 or H460 cells were transfected with a pGL4-luc2 (Promega) construct to stably express the luciferase gene, selected using 100 μg/mL of Neomycin (Sigma-Aldrich) and pooled for in vivo experiments. siRNA, GpR sequences and plasmids are listed in Supplementary Table 1 and Supplementary Table 6.

**Transcriptome analysis.** RNA-seq reads were quality checked with FastQC (https://www.bioinformatics.babraham.ac.uk/projects/fastqc/) and aligned in paired-end mode to the human genome assembly (GRCh37) using the RSubread package aligner[61] with the default settings. Mapped data were converted to gene level integer read counts using feature Counts (RSubread package) and the Ensemble GTF annotation (Homo_sapiens.GRCh37.74). Expression of a gene was measured in RPKM (Reads Per Kilobase Million) units.

**Differentially expressed gene (DEG) analysis.** Differential expression (DE) was evaluated comparing the gene level integer read count data for the knockdown and control samples using the DESeq2 Bioconductor package[62] with default settings. A gene was considered differentially expressed if its abundance changed more than 1.5-fold between the two conditions. The resulting p values were adjusted (padj) using the Benjamini and Hochberg approach for controlling the false discovery rate (FDR). Genes with an adjusted p value determined to be <0.05 (FDR < 0.05) by DESeq2 with a fold change value ≥1.5 or ≤0.8 between two groups were considered to be differentially expressed.

**Network analysis.** The integrative analysis was performed using data from KIMAT1 KD or DHX9/NPM1 KD RNA-seq to find shared enriched pathways within the sets of differentially expressed (DE) genes which are predicted targets of a pre-defined set of miRNAs. GSEA was performed and a consensus set of pathways were developed into a network diagram. Predicted miRNA targets for the set of miRNAs were acquired from the Targetscan database (release 7.2). Predicted targets which were DE within each siRNA experiment were subject to GSEA using the Hallmark and C6 Oncogenic gene sets (MSigDB version 7); this was performed using the GSEA_r package (version 1.2) within the R environment (version 3.6.1). Gene sets (pathways) which reached significance (FDR < 0.05) were compared between the experiments and a set was chosen to produce a network graph. The network graph displays miRNAs (blue squares) and pathways (orange circles) and the connections between them (purpledashed: DHX9; green-solid: NPM1; gray solid: KIMAT1); edge-weights represent the number of DEGs contributing to the connection within each gene set. The network was produced using the igraph R package (version 1.2.4.2).

**MiRNA sequencing.** Small RNA-seq raw reads from control and treated (KIMAT1 KD) samples were quality checked using FastQC (https://www.bioinformatics.babraham.ac.uk/projects/fastqc/). The FASTX-Toolkit (http://hannonlab.cshl.edu/fastx_toolkit/) was used to trim 3' small RNA-seq adapter sequence. Reads smaller than 18 nt were discarded. The cleaned reads for each sample along with the human reference genome (hg19), all the known human hairpins and human mature miRNA sequences from miRBase database (Release 21) were used as input in mirDeep2 to quantify miRNA abundances. The raw miRNA counts form mirDeep2 were used in DESeq2[62] to identify differentially expressed miRNAs between control and treated samples at a FDR threshold of ≤0.05.

**Identification of lung cancer-associated protein-coding genes and lncRNAs.** Differentially gene expression datasets were compiled by the Molecular Biology Core Facility Team (MBCF) at Cancer Research UK Manchester Institute. The Cancer Genome Atlas (TCGA), Lung Adenocarcinoma (LUAD) and Lung Squamous Cell Carcinoma (LUSC) alignment files (BAM) were downloaded from dbGap and processed to generate raw expression files. We extracted gene expression matrices specifically for cancer patients which were composed of 542 and 502 samples respectively. Differential expression between these cancer patients against 427 normal lung samples taken from Genotype-Tissue Expression (GTex) (version 7) was performed. Only genes that were present and had a minimum count of 11 across all samples in their respective comparisons were considered for differential expression analysis. R Bioconductor package DESeq2 (v1.26.0) was used and a gene was considered differentially expressed if it passed a false discovery rate threshold of 5%. Scatterplots were generated using the overlapped differentially expressed genes between NGS experiments and TCGA/GTex.

**Expression of KIMAT1 in TCGA, GTEx and cell lines from CCLE.** TCGA data for a total of 10,480 tumors across 33 cancer types were acquired from the Cancer Genomics Cloud. Gene counts for GTEX samples (n = 11,688) were downloaded from https://gtexportal.org/home/. The raw read counts were first converted to RPKM values and then transformed to log2 scale. A pseudo-count of 1 was added to avoid taking log of zero. Samples on x axis are ordered by the average expression of the lncRNAs. KIMAT1 expression data were obtained from the file CCLE_RNAseq_081117.rpkm.gct downloaded from CCLE data portal (http://portals.broadinstitute.org/ccle/data). Cell lines of interest were grouped by their respective tissue types. The expression data, in the form of RPKM values, were converted to log2 scale and a pseudo-count of 1 was added to avoid taking log of zero. A total of 863 cell lines across 19 tissue types were analyzed.

**CCLE and TCGA KRAS copy number.** Raw Affymetrix CEL files were converted to a single value for each probe set representing a SNP allele or a copy number probe. Copy numbers were then inferred based upon estimating probe set specific linear calibration curves, followed by normalization by the most similar HapMap normal samples. Segmentation of normalized $\log_2$ ratios (specifically, $\log_2 (CN/2)$) was performed using the circular binary segmentation (CBS) algorithm. TCGA copy number annotation: Copy number values: $-2$ = homozygous deletion; $-1$ = hemizygous deletion; 0 = neutral/no change; 1 = gain; 2 or more = high level amplification.

**Overall survival analysis.** Disease free survival and overall survival (OS) analysis were performed on LUAD and LUSC datasets considering KRAS amplification plus mutation. Kaplan–Meier curves are generated at 2.5 years, 5 years and 20 years follow-up period using samples for which both survival and expression data were available. BAM files for all these datasets were downloaded from GDC (https://portal.gdc.cancer.gov/) and counts were extracted from the BAM files using feature Counts.

**Gene Ontology (GO) and gene set enrichment analysis (GSEA).** Gene ontology and gene set enrichment analysis were performed to investigate whether the differentially expressed genes (DEGs) were part of specific pathways. The GO functional annotation was explored with BINGO. Gene signature Analysis was based on MSigDB v6.2 database using the GSEA desktop implementation software[63]. Genes

in the dataset were pre-ranked and weighted by the independent gene-level Wald statistics and 1000 phenotype-based permutations were conducted. To understand the functions of *KIMAT1* regulated genes GSEA was performed using the HALLMARK and C6 oncogenic signatures and a gene $\log_2$-fold change. Significant terms with $p$ values less than 0.05 were considered significant.

**Cross-linked RNA Immunoprecipitation (CLIP)**. CLIP was performed as previously described[64]. Cells were UV irradiated at 0.8 J/cm², lysed in RIPA buffer (Sigma-Aldrich) with 1× protease inhibitor cocktail (Sigma-Aldrich) and RNase inhibitor (NEB) for 10 min in ice. Lysates were then precleared with Protein G beads (Thermofisher Scientific) for 1 h at 4 °C, and immunoprecipitated with IgG or the indicated antibodies for 3 h at 4 °C. Immuno-complexes were precipitated with Protein G beads and washed six times with washing buffer (50 mM Tris-HCl pH 7.5, 150 mM NaCl, 1 mM MgCl₂, and 0.05% IGEPAL CA-630). 10% of beads were boiled with 1× Laemli buffer (Bio-Rad) at 95 °C for 5 min and loaded on a polyacrylamide gel to verify immunoprecipitation efficiency. The remaining beads were treated with TurboDNase (Thermofisher Scientific) and Proteinase K (NEB) and RNA was isolated using the TRIzol solution. UBC was used as a negative control.

**Native RNA pull-down assay**. DNA of *KIMAT1* full length or deletion constructs were amplified by PCR. T7 RNA polymerase promoter sequences were added to forward primers for subsequent in vitro transcription. PCR products were purified with Gel Extraction Kit (Qiagen) and transcribed in vitro using Biotin RNA labeling Mix Kit (Roche) and T7 RNA Transcription Polymerase (Roche) according to the manufacture's instruction. 3 μg of Biotin-labeled lncRNA (full-length or fragments) were diluted into 40 μl RNA structure buffer (10 mM Tris-HCl pH7.0, 10 mM MgCl₂, 0.1 M KCl), heated at 90 °C for 2 min and placed in ice for additional 2 min. Samples were kept at room temperature for 30 min to generate RNA secondary structures. Meanwhile, 80% confluent cells were washed with cold PBS and collected. After centrifugation, cells were re-suspended in 1 ml chilled RIP buffer (25 mM Tris-HCl pH7.4, 150 mM KCl, 5 mM EDTA, 0.5% NP40) and sheared using a Bioruptor device with 30 strokes of 30 s on and 45 s off. 2 mg of sheared lysates were added to the folded RNA in RIP buffer supplemented with a final concentration of 0.1 μg/μl tRNA at 4 °C for 2 h. Pre-washed streptavidin beads were mixed with RNA-cell lysate complex and further incubated for 1 h at 4 °C. Beads were washed six times and boiled in 1× Laemmli loading buffer (Bio-Rad). Retrieved protein samples were examined using immunoblotting.

**Co-immunoprecipitation (Co-IP)**. $5.0 \times 10^7$ cells were washed with pre-chilled PBS and re-suspended in 400 μl RIPA buffer (Sigma-Aldrich) plus protease inhibitors (Sigma-Aldrich). Pre-cleared lysates were immunoprecipitated with the indicated antibodies for 3 h at 4 °C. Next, the immunocomplex was incubated with Protein G beads overnight at 4 °C, washed six times with washing buffer (50 mM Tris-HCl pH 7.5, 150 mM NaCl, 1 mM MgCl₂, and 0.05% IGEPAL CA-630), boiled with Laemli buffer (Bio-Rad) at 95 °C for 5 min and analyzed by immunoblotting.

**Ubiquitination assay**. H1299 cell were transfected with GpRs Ctrl or GpRs *KIMAT1* for 48 h and treated with MG132 (20 μM) for 4 h. Then, cells were harvested and lysates incubated with DHX9 or NPM1 antibody plus Protein G beads overnight at 4 °C. The precipitated complex was boiled in 1× Laemli buffer at 95° and the supernatant was used for immunoblotting with an ubiquitin antibody.

***KIMAT1* transcript boundaries**. *KIMAT1* transcription starting site (TSS) was identified using CAGE-seq counts data obtained from gastric and lung adenocarcinoma cell lines from the FANTOM5 study (https://fantom.gsc.riken.jp/5/). JASPAR was used to predict the likelihood of transcription factors binding sites in *KIMAT1* sequence.

**Cell viability and IncuCyte cell growth analysis**. $5.0 \times 10^3$ cells were cultured in 96-well plates. Cell viability was assessed using the CellTiter 96 Aqueous One Solution Cell Viability assay (Promega) and measured at 490 nm as per manufacturer's instructions in a Multilabel Counter (SpectraMax M5). Cell Confluence was analyzed using the IncuCyte Zoom live-cell imaging systems from Essen Bioscience. Cells seeded into a 96-well plate were transfected with 50 nM of GapmeRs using Lipofectamine RNAiMAX. Phase-contrast images were then taken every two hours for a total of 72 h and the percentage of confluence was calculated with the Incucyte Zoom software.

**Caspase-Glo 3/7 and Annexin V assay**. $5.0 \times 10^3$ cells were seeded in 96-well plates for 48 h. Caspase-3/7 activity was measured by adding 100 μl of Caspase-Glo 3/7 solution (Biorad) for 30 min at room temperature in the dark. For the Annexin V assay, cells were grown in 6-well plates, transfected with GpRs for 48 h and then washed with cold PBS and harvested with trypsin. Cell pellets were incubated with Annexin V for 15 min (Trevigen) in the dark at room temperature. 400 μl 1× binding buffer was then added to the cells and the percentage of apoptotic cells analyzed using Flow Cytometry (NovoCyte NovoExpress Software version 1.3.0).

**Colony formation assay and cell count**. $5.0 \times 10^3$ cells were seeded in six-well plates. 2 weeks later cells were washed with PBS and fixed with cold methanol, stained with 0.05% crystal violet (Sigma-Aldrich), photographed and counted using the GelCount System (Oxford Optronix) and the GelCount™ operating software.

**3D proliferation assay**. $2.0 \times 10^3$ cells either transfected with *KIMAT1* GpRs or DHX9 and NPM1 siRNAs, DHX9 KO and NPM1 KO cells or with stable over-expression of *KIMAT1*, were cultured on ultra-low attachment plates (ULA, Corning) and incubated at 37 °C for 10–15 days. Tumorsphere area was analyzed using Image J software.

**3D invasion assay**. $2.0 \times 10^3$ cells in 200 μl RPMI-1640 were placed in ULA plates and incubated at 37 °C for 7 days. Next, 100 μl of medium was gently removed and 100 μl of RPMI-1640 medium containing 3.8 μg/ml of Matrigel (Corning) was added into the wells. Cells were kept at 37 °C for 7 more days. Area of the tumorspheres was quantified with Image J software.

**Single-molecule fluorescence in situ hybridization (smFISH)**. To detect *KIMAT1* at the single cell level a pool of 27 custom Stellaris FISH probes were designed using the online tool (https://www.biosearchtech.com) to sequentially cover *KIMAT1* RNA sequence. To avoid off-target hybridization, BLAST was used to remove unspecific probes. Probes were labeled with CAL Fluor Red 590 or Quasar 570 dye and synthesized by LGC Biosearch Technologies. Cells were grown on a coverslip for 48 h and fixed with 4% formamide at room temperature for 10 min, washed with PBS and permeablized with 70% ethanol for one hour at 4 °C. Then, cells were incubated in the dark with the hybridization buffer (LGC Biosearch Technologies) containing the probes (125 nM) at 37 °C for at least 4 h and counterstained with DAPI (5 ng/ml) at 37 °C for 30 min. Images were acquired with gSTED microscope and spots were counted using the online JAVA software of StarSearch[57]. FISH probes are listed in Supplementary Table 2.

**RNA antisense purification and mass spectrometry**. RAP-MS was performed as previously described[21]. To pull down endogenous *KIMAT1*, 5' biotinylated 20-mer antisense oligonucleotides were designed using the online tool (https://www.biosearchtech.com) and synthesized by LGC Biosearch Technologies. Cells (2.0 × $10^8$ cells per sample) were UV cross-linked and lysed in 2 ml RIPA buffer (SigmaAldrich). Cell lysates were sonicated using a Bioruptor shearing device. Precleared cell lysates were mixed with 20 μg of lncRNA antisense probes and incubated for 2 h at 67 °C with intermittent mixing at $1100 \times g$ on a thermomixer. Washed beads were added to the lysate-probes complex and further incubated for 2 h at 67 °C. Beads were washed six times and boiled in 1 x Laemli loading buffer (Bio-Rad). Protein lysates were loaded on a SDS-PAGE gel (Bio-Rad) and protein bands were destained in 1 ml of 100 mM ammonium bicarbonate, 40% methanol at 37°C. Gel bands were then washed in 1 ml of HPLC grade water for 10 min before removal of the water. Gel pieces were dehydrated by the addition of 1 ml acetonitrile for 10 min followed by the removal of the acetonitrile. The water-acetonitrile hydration-dehydration cycle was repeated a total of 3 times. Dehydrated gel pieces were then rehydrated in 100 μl of 40 mM ammonium bicarbonate, 10% acetonitrile carrying 12.5 ng/μl sequencing grade trypsin for 20 min before the removal of any excess. Gel pieces were then covered with 100 μl of 40 mM ammonium bicarbonate, 10% acetonitrile and incubated at 37 °C for 16 h. Digests were acidified by the addition of trifluoroacetic acid to a final concentration of 0.1% before removal of the digest from the gel with subsequent drying of the sample in a vacuum centrifuge. Dried peptide samples were then resuspended in 2% acetonitrile, 0.05% trifluoroacetic acid prior to injection on the nano LC system Peptides were injected directly onto a 25 cm long 75 μm ID, 2 μm C18 pepmap EasySpray column (Thermo) using an RSLCn HPLC system (Thermo) at a flow rate of 200 nl/min. Peptides were separated with a gradient of 1–22% acetonitrile 0.1% formic acid over 30 min at a temperature of 60 °C. The EasySpray column interfaced directly into an Orbitrap Fusion mass spectrometer with a spray voltage of 1.4KV. The instrument was operated in data-dependent mode with a 120,000 resolution orbitrap MS1 scan over a mass range of 350–1000 m/z with a target value of 2E5 ions and a maximum fill time of 50 ms. MS2 HCD spectra were collected in the linear ion trap with a 3 m/z isolation window, 28% normalized collision energy, rapid scan rate with a target value of 1e4 ions and a maximum fill time of 50 ms. Raw data was processed using Mascot Distiller (Matrix Science) to generate MGF peak lists which were subsequently submitted to a Mascot (Matrix science) database search. Mascot search results were then imported in Scaffold 4 (Proteome software). Two independent biological replicates were analyzed and to minimize the potential background two separate samples, as negative control, were incubated either with the housekeeping gene *Ubiquitin C* (UBC) or treated with RNase (10 ug/ml at 37 °C for 30 min) prior the hybridization step. 5' biotinylated 20-mer antisense oligonucleotides are listed in Supplementary Table 2. The mass spectrometry data have been deposited to the ProteomeXchange Consortium via the PRIDE[65] partner repository with the dataset identifier PXD024388 and 10.6019/PXD024388".

**Sequential immunofluorescence and FISH assay**. Cells were grown on coverslips for 48 h and fixed with 4% PFA at room temperature for 10 min, permeablized with 0.2% Triton X-100/PBS for 5 min at room temperature. Coverslips were then

incubated with anti-DHX9 or anti-NPM1 primary antibodies and Alexa Fluor 488-conjugated secondary antibodies and re-fixed with 4% PFA. smFISH was performed as previously described. Digital photographs were acquired with a Two-Photon Excitation gSTED Microscope (Leica) and visualized in Leica Advanced Fluorescence software (Leica).

**Statistics and reproducibility**. Error bars in all the plots indicate mean ± S.D. $p$ value < 0.05 was considered statistically significant. **$p$ value < 0.001, *$p$ value < 0.05 by two tailed Student's $t$ test. All experiments were performed at least three times unless otherwise indicated. Statistics was calculated with Excel 2010 or GraphPad Prism 8.

**Reporting summary**. Further information on research design is available in the Nature Research Reporting Summary linked to this article.

## Data availability
RNA-seq and ChIP-seq data generated in this study have been deposited in the Gene Expression Omnibus database under the accession code: GSE124631. The mass spectrometry data have been deposited to the ProteomeXchange Consortium via the PRIDE partner repository with the dataset identifier PXD024388 and https://doi.org/10.6019/PXD024388". A list of proteins interacting with KIMAT1 by mass spectrometry is provided in Supplementary Data 3. Publicly available ChIP-seq data are available from ENCODE (https://www.encodeproject.org). Some of the data/analyses presented in the current publication are based on the use of study data downloaded from the dbGaP web site, under phs000178.v11.p8/ https://www.ncbi.nlm.nih.gov/projects/gap/cgi-bin/study.cgi?study_id=phs000178.v11.p8. Source data are available as a Source Data file. The remaining data are available within the Article, Supplementary Information or available from the authors upon request. Source data are provided with this paper.

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

## Acknowledgements

We would like to thank J. Downward for the kind gift of Type II pneumocytes cells and F. Trapani for the independent evaluation of the IHC slides in the animal studies. We are grateful to L. Doar of the CRUK Biological Resources Unit for the help with the in vivo studies. Work in M.G. lab is funded by CRUK core grant (C5759/A20971) and Lung Cancer Centre (C5759/A20465).

## Author contributions

L.S. and M.G. designed the project. L.S., P.M., T.M. A.R.P. performed in vitro studies. L.S. performed the in vivo studies. L.B.S. and S.C. performed the PDX study. S.S. led the bioinformatics analysis with H.S.L., D.L. and R.S. S.V. helped with bioinformatics analysis. M.F. and F.G. performed histopathological analysis. D.D.S. performed Massspectrometry analysis. K.Z. helped with microscopy and acquired smFISH and immunofluorescence images. G.D.L. provided thoughtful suggestions and helped in editing the manuscript. R.O.'-K. performed miRNA in vitro processing. L.S. and M.G. wrote the manuscript. M.G. supervised all research.

## Competing interests

M.G. and L.S. filled the patent application PCT/GB2020/052449, "Identification of KRAS-responsive lncRNAs". The other authors declare no competing interests.
