## [Peer Review File · Nature Communications]

Reviewer #1 (Remarks to the Author): Expert in lncRNA computational

In this manuscript, the authors identified a KRAS (both Mut and WT) responsive lncRNA located at the nucleus and cytoplasm. The expression of KIMAT1 and KRAS upregulated in the advanced stage of lung adenocarcinoma. KIMAT1 expression shows a strong positive correlation with KRAS. Myc is a downstream target of KRAS and a transcription factor. It binds to the KIMAT1 promoter MER101 LTRs region. This revealed that lncRNA KIMAT1 was regulated by transcription factor Myc. The authors hypothesize that KIMAT1 promotes lung cancer cell growth, cell survival, and invasion ability through interacting with two key proteins DHX9 and NPM1, which are two components of microprocessor complex (MC) at 5' end 399 nt region and 3' end 258 nt region respectively. KIMAT1 regulated the stability of these two proteins by decreasing the polyubiquitination and degradation of these two proteins through the proteasome. KIMAT1 and its interacting proteins regulated KRAS signaling included KRAS target genes, and KIMAT1 KD was associated with EMT, which implied that KRAS responsive lncRNA KIMAT1 might regulate cancer cell metastasis through EMT. DHX9 (RNA helicase A) interacts with DDX5 (another type of RNA helicase). NPM1 also interacts DDX5 and all these proteins, including Drosha, are key components of MC that mainly regulate miRNAs biogenesis and processing from pri-miRNAs. The authors identified two groups of miRNAs that were regulated by KIMAT1 and its interacting proteins. One group included oncogenetic miRNAs and another group included tumor-suppressive miRNAs. MC components DHX9, NPM1, DDX5 and Drosha directly bind with oncogenetic pri-miRNAs and increased these miRNAs processing; DHX9, NPM1 and DDX5 could not directly bind with tumor-suppressive pri-miRNAs but could still regulate these miRNAs expression. p21 as a tumor suppressor was found upregulated when knockdown KIMAT1. p21 served as a novel component of the MC and directly interact with Drosha, together bind with tumor-suppressive pri-miRNAs and upregulate the expression of these miRNAs. p21 at the same time abolished the interaction between DHX9, DDX5 and NPM1, DDX5. p21 also downregulated the protein expression level of DDX5, DHX9, and NPM1 which led to downregulation of oncogenetic miRNAs by interfering the binding among MC components. Finally, in vivo studies demonstrated that knockdown of KIMAT1 decreased the PDX model tumor volume, downregulated cell proliferations and upregulated cell apoptosis. Overexpression of KIMAT1 led to increased metastasis of lung cancer cells to the liver and kidney. NSG mice injected with DHX9 or NPM1 KO cells decreased KRAS expression. Although interesting, this reviewer has several concerns that need to be addressed.

Major Concerns:

1. The knockdown effect of lncRNA KIMAT1 is well demonstrated, but the overexpression effect of the KIMAT1 is not clear. Also, the structure and isoforms of KIMAT1 are not shown.
2. In Figure 2a, CAGE-seq analysis used lung adenocarcinoma cell line IA-LM. However, in Figure 2b, ChIP-seq analysis is used H1299 and A549 cell lines. H1299 is the primary cell line used in this manuscript. To be consistent, CAGE-seq should also show in the H1299 cell line.
3. In Figure 2, the evidence showing Myc is the transcription factor that regulates the expression of KIMAT1 is not enough, ChIP-qPCR may also need to be performed and confirmed the binding of Myc at the promoter region of KIMAT1.
4. All the colony formation assay figures are not clear. The resolution needs to be adjusted.
5. Figure 4J, NPM1 KO in two trials are not complete. There are quite some differences between KO-1 and KO-2 knockout efficiency. However, in Figure 4K, NPM1 KO 3D invasion assay showed no difference between KO-1 and KO-2. DHX9 KO-1 and KO-2 are complete, but 3D invasion assay showed a significant difference between KO-1 and KO-2. What could be the explanation of these discrepancies?
6. Supplementary Figure 4k, western blot of polyubiquitination did not show the ubiquitination site. In addition, the ubiquitin of DHX9 IP trial do not see any difference when KD KIMAT1. Authors need more solid evidence to show KIMAT1 regulates NPM1 and DHX9 stability by decreasing the protein degradation by the proteasome, such as polyubiquitination assay.
7. In Figure 5e, since the major cancer type the manuscript is talking about is lung adenocarcinoma, why the analysis used KRAS kidney instead of KRAS lung?
8. According to the hypothesis, E2F, KRAS and EMT should be seen in common among KIMAT1 KD, DHX9 KD and NPM1 KD. However, only KRAS kidney showed the three KDs in common. KRAS lung only shows a downregulation when knockdown NPM1. Also, KRAS downstream target RAF and MEK do not have three KDs in common either. The authors should explain this.
9. DHX9 and NPM1 overexpression models may need to be included during analysis or performing qRT-PCR showing the changing of KRAS target genes after overexpressing DHX9 and NPM1. There

is one figure in Supplementary Figure 5 showed the EMT marker but only one marker. There are many EMT markers and KRAS downstream targets (e.g., RAF, MEK and ERK) need to be detected.

10. Can these markers be added into the measurements when demonstrating KIMAT1 and its interacting proteins regulate EMT and KRAS signaling?
11. From Figure 6f, the western blot apparently showed that KIMAT1 not only affects the stability of DHX9 and DDX5 but also interaction among them. Moreover, Figure 6d showed the binding between DHX9 and NPM1, is this interaction mediated by KIMAT1 or these two proteins in the nucleus can interact with each other? The role of KIMAT1 in DHX9, NPM1, DDX5 and the interaction among these three proteins need to be demonstrated/discussed in more detail.
12. Supplementary Figure 6d and f showed DHX9 and NPM1 do not directly bind to miR-200, 27, 7, 139 but can still regulate these miRNAs processing, which showed in Supplementary Figure 6b, c. What could be the potential mechanism to explain this? Or is this due to the decreasing of p21 expression? Then how KIMAT1, DHX9, and NPM1 regulate p21 expression?
13. Figure 7e, f showed p21 affects the binding between DHX9, DDX5, and the binding between DDX5 and NPM1. At the same time, p21 overexpression downregulated the protein expression of these three proteins by post-transcriptional regulation. What could be the potential explanation or mechanism? How p21 regulate these three proteins expression level and how KIMAT1 regulates p21?
14. Figure 8c showed that KIMAT1 KD led to increasing in KRAS. This is the opposite of the mechanism and what the text was described. Why?
15. KIMAT1 KD mouse model should be included to addressing the metastasis ability.
16. Authors should demonstrate in the mouse model that if DHX9 or NPM1 KO can abolish the metastasis function of KIMAT1 in KIMAT1 overexpression cells.

Minor Concerns:

1. Figure legends such as Figure 4d, 4j is not clear. The labeling of the figures such as Figure 3d, 3f, the cell line labeling is confusing.
2. It is better if the authors could include the Xenograft model of KIMAT1 KD by treating LNA-GpR in Figure 3.
3. In Figure 4h, j, the IP efficiency should be shown by western blot.
4. In several qRT-PCR results, the statistical significance labels are not clear. There was more than one comparison between bars, but there is only one significance label showing in the figures. The authors should fix this problem.
5. Western Blot in Figure 6c, e should show a longer time exposure blot.
6. In Figure 7b, statistical significance could be shown.
7. In Figure 8f, mice bearing H460 cells, which is a KRAS Mut metastatic cell line. However, after the growth of 4 weeks, neither the EV treatment or the KIMAT1 treatment showed metastasis to other organ such as liver or kidney. Why?

Reviewer #2 (Remarks to the Author): Expert in lncRNA

Shi and colleagues describe a new KRAS-responsive lncRNA that has important roles in driving tumorigenesis. A model is proposed in which the lncRNA promotes processing of oncogenic miRNAs and halts biogenesis of miRNAs with tumor suppressor function. There is a lot of data in the paper and it is generally clear. It should be of broad interest. The authors should address the following points:

(1) The authors have used CAGE to map the 5' end of the lncRNA, but it does not appear they have validated the exact sequence of the mature transcript. Where is the 3' end? Is it spliced as the annotation suggests? This is important to ensure that the overexpression plasmids are generating the physiologically relevant transcript. The authors should also mention if it is conserved in any other species.

(2) Fig 3: It would be really nice to show that the authors can rescue some of the gapmer effects with a lncRNA expression plasmid that is resistant to the LNA-gapmer. This would prove without any doubt that the effects are "on-target".

(3) Fig 6b: Can the authors speculate on why the mature and precursor data do not correlate very

strongly in many cases? For example, for 200a, the precursor goes up 5 fold with gapmer #1, but the mature is only slightly increased.

(4) It would be helpful in the discussion to talk about how the authors envision DHX9/NPM1 only regulates a subset of pri-miRNAs.

(5) Drosha processing occurs in the nucleus but most of the lncRNA is in the cytoplasm, so there is a bit of a disconnect here that the authors should discuss more in the text.

Minor points

(1) P.3: Some words are missing in the sentence that begins "miRNAs bind to the 3'UTR of target genes." I believe the authors meant to say that miRNAs can induce mRNA degradation.

(2) I see that a few papers refer to the Microprocessor complex as MC, but this does not appear to be very standard in the field.

(3) P.4: For a general audience, please explain what the difference between "amplification of the KRAS gene" and "copy number gain of the KRAS gene." They sound like the same thing to me.

(4) P.5: Supp Fig 1h is called out in the wrong place in the text.

(5) Supp Fig 1l: Please include EV and KRAS labels on x-axis.

(6) Fig 2d: The x-axis could be labeled more simply.

(7) Supp Fig 4h: It looks like GAPDH levels also increase so this is not a good control. Quantification of blots in Supp Fig 4h-k would be helpful.

(8) P.9: "To define the transcriptome modulated by KIMAT1 and its interacting proteins we used RNA-seq": RNA-seq does not reveal interacting proteins.

(9) Fig 5d: It would be more helpful to have all 3 datasets together rather than separated out as currently displayed.

(10) Fig 6c: It would be good to use a different miRNA that is not affected in cells by KIMAT1 KD and show that its in vitro processing is not altered.

(11) P.12: The data in Fig 6f is interpreted too strongly in the main text. The RNase A treatment data could support KIMAT1 being involved, but it is also consistent with many other RNAs being involved.

Reviewer #3 (Remarks to the Author): Expert in mouse models of lung cancer

The authors in this manuscript describe the identification of a KRAS-responsive long non-coding RNA that controls microRNA processing and regulate tumorigenesis. They focused on the Wild-type KRAS (KRASWT) amplification and investigated the role of KRASWT overexpression in lung cancer progression. The authors identified and characterized a KRAS-responsive lncRNA, KIMAT1 and showed that it correlates with KRAS levels both in cell lines and in lung cancer specimens. They also explored the mechanisms to show that KIMAT1 is a MYC target and drives lung tumorigenesis by promoting the processing of oncogenic miRNAs through DHX9 and NPM1 stabilization while halting the biogenesis of miRNAs with tumor suppressor function via MYC-dependent silencing of p21. They conclude that KIMAT1 in maintaining a positive feedback loop that sustains KRAS signaling during lung cancer progression.

The manuscript may be intriguing. However, some of the points and experimental data are confusing especially in some animal studies.

1. The authors used TCGA data to show that high KRAS expression or KRAS amplification had a poorer overall survival (OS) or disease-free survival compared to patients with low KRAS expression. It is confusing that as mentioned that a total of 34% of lung adenocarcinoma (LUAD) and 48.8% of lung squamous cell carcinoma (LUSC), with consequent increase of KRAS mRNA expression (Fig. 1a and Supplementary Fig. 1a). 17% of adenocarcinoma patients with KRAS gain/amplification also harboured a mutant KRAS allele (Supplementary Fig. 1b and Supplementary Table 1).

However, Fig 1b. in LUAD the author used KRAS high vs KRAS low, and in LUSC, the author used KRAS amplified vs not amplified and in amplified group, only 10 cases were included. The reader may be confused why the author use different criteria and what are the rationale? The number of KRAS amplified is also too small to draw any conclusion.

2. The reader may be interested to know what is the difference in the regulatory mechanism of this pathway between KRAS mutant vs KRASWT with KRAS amplification?

3. The author showed that MYC and regulate KIMT1 expression, but it is not clear whether KRAS regulate KIMT1 is MYC dependent.

4. As the author mentioned that KIMAT1 in maintaining a positive feedback loop that sustains KRAS signaling and KIMAT1 KD induces MYC downregulation. However, in Fig. 8h, the feed back regulatory mechanism was not seen.

5. The animal study in Fig. 8 b,d,e,g and Supplementary 8 c, f, g are confusing, the author used arrow to indicate the tumors, however the readers may not be convinced that they are really metastatic tumors. The resolutions are not adequate, lower power field pictures are not shown to tell the difference of tumor nodules and the arrow heads are too big, even bigger than the xenograft tumor nodules.

6. In p 15, a patient with limp nodes metastasis, should it be with lymph nodes metastasis?

REVIEWER COMMENTS

Reviewer #1 (Remarks to the Author): Expert in lncRNA computational

In this manuscript, the authors identified a KRAS (both Mut and WT) responsive lncRNA located at the nucleus and cytoplasm. The expression of KIMAT1 and KRAS upregulated in the advanced stage of lung adenocarcinoma. KIMAT1 expression shows a strong positive correlation with KRAS. Myc is a downstream target of KRAS and a transcription factor. It binds to the KIMAT1 promoter MER101 LTRs region. This revealed that lncRNA KIMAT1 was regulated by transcription factor Myc. The authors hypothesize that KIMAT1 promotes lung cancer cell growth, cell survival, and invasion ability through interacting with two key proteins DHX9 and NPM1, which are two components of microprocessor complex (MC) at 5' end 399 nt region and 3' end 258 nt region respectively. KIMAT1 regulated the stability of these two proteins by decreasing the polyubiquitination and degradation of these two proteins through the proteasome. KIMAT1 and its interacting proteins regulated KRAS signaling included KRAS target genes, and KIMAT1 KD was associated with EMT, which implied that KRAS responsive lncRNA KIMAT1 might regulate cancer cell metastasis through EMT. DHX9 (RNA helicase A) interacts with DDX5 (another type of RNA helicase). NPM1 also interacts DDX5 and all these proteins, including Drosha, are key components of MC that mainly regulate miRNAs biogenesis and processing from pri-miRNAs. The authors identified two groups of miRNAs that were regulated by KIMAT1 and its interacting proteins. One group included oncogenetic miRNAs and another group included tumor-suppressive miRNAs. MC components DHX9, NPM1, DDX5 and Drosha directly bind with oncogenetic pri-miRNAs and increased these miRNAs processing; DHX9, NPM1 and DDX5 could not directly bind with tumor-suppressive pri-miRNAs but could still regulate these miRNAs expression. p21 as a tumor suppressor was found upregulated when knockdown KIMAT1. p21 served as a novel component of the MC and directly interact with Drosha, together bind with tumor-suppressive pri-miRNAs and upregulate the expression of these miRNAs. p21 at the same time abolished the interaction between DHX9, DDX5 and NPM1, DDX5. p21 also downregulated the protein expression level of DDX5, DHX9, and NPM1 which led to downregulation of oncogenetic miRNAs by interfering the binding among MC components. Finally, in vivo studies demonstrated that knockdown of KIMAT1 decreased the PDX model tumor volume, downregulated cell proliferations and upregulated cell apoptosis. Overexpression of KIMAT1 led to increased metastasis of lung cancer cells to the liver and kidney. NSG mice injected with DHX9 or NPM1 KO cells decreased KRAS expression. Although interesting, this reviewer has several concerns that need to be addressed.

Major Concerns:

1. The knockdown effect of lncRNA KIMAT1 is well demonstrated, but the overexpression effect of the KIMAT1 is not clear. Also, the structure and isoforms of KIMAT1 are not shown.

Author's reply:

- We stably overexpressed KIMAT1 in H1299 and H460 cells and confirmed its overexpression by qPCR (Supplementary Fig. 5a). In addition, we showed that KIMAT1 overexpression promoted 3D invasion (Fig. 3f), colony formation (Supplementary Fig. 5b) and tumor growth when cells stably expressing KIMAT1 were injected subcutaneously in nude mice (Fig. 3g, 3h and Supplementary Figure 5c).

- There is only one KIMAT1 isoform as reported in Ensembl (Supplementary Fig. 1e). The secondary structure of KIMAT1 has been added in Supplementary. Fig. 1g.
2. In Figure 2a, CAGE-seq analysis used lung adenocarcinoma cell line IA-LM. However, in Figure 2b, ChIP-seq analysis is used H1299 and A549 cell lines. H1299 is the primary cell line used in this manuscript. To be consistent, CAGE-seq should also show in the H1299 cell line.
- As suggested by the reviewer, CAGE-seq analysis has been performed in H1299 and is reported in Fig. 2a. The data was obtained from the manuscript *Nat Genet. 2017 Jul;49(7):1052-1060. doi: 10.1038/ng.3889.*
3. In Figure 2, the evidence showing Myc is the transcription factor that regulates the expression of KIMAT1 is not enough, ChIP-qPCR may also need to be performed and confirmed the binding of Myc at the promoter region of KIMAT1.
- ChIP-qPCR has been performed and is reported in Fig. 2e. We used TFAP4, a known MYC target, as positive control.
4. All the colony formation assay figures are not clear. The resolution needs to be adjusted.
- To improve the resolution we reacquired the images or repeated the colony assays in Supplementary Fig. 2a, 4c, 4g, 5b, 5f, 6c, 6f, 13b.
5. Figure 4J, NPM1 KO in two trials are not complete. There are quite some differences between KO-1 and KO-2 knockout efficiency. However, in Figure 4K, NPM1 KO 3D invasion assay showed no difference between KO-1 and KO-2. DHX9 KO-1 and KO-2 are complete, but 3D invasion assay showed a significant difference between KO-1 and KO-2. What could be the explanation of these discrepancies?
- We generated several DHX9 KO clones and the discrepancy could be due to the fact that different clones were used for the western blot and 3D cell invasion assay. We have repeated these experiments using the same DHX9 KO clones in Fig. 4j and Fig. 4k and there is clearly a strong correlation between DHX9 KO and 3D cell invasion.
6. Supplementary Figure 4k, western blot of polyubiquitination did not show the ubiquitination site. In addition, the ubiquitin of DHX9 IP trial do not see any difference when KD KIMAT1. Authors need more solid evidence to show KIMAT1 regulates NPM1 and DHX9 stability by decreasing the protein degradation by the proteasome, such as polyubiquitination assay.
- Thanks for your suggestion. We repeated this experiment using a different protocol and a new ubiquitin antibody as described in *Jin L, et al. Long noncoding RNA MEG3 regulates LATS2 by promoting the ubiquitination of EZH2 and inhibits proliferation and invasion in gallbladder cancer. Cell Death Dis 9, 1017 (2018).* The new polyubiquitination assay is reported in the Methods at page 30. The results are reported in Supplementary Fig. 7d. From

this experiment it is clear that in the absence of *KIMAT1*, *DHX9* and *NPM1* are ubiquitinated.

7. In Figure 5e, since the major cancer type the manuscript is talking about is lung adenocarcinoma, why the analysis used KRAS kidney instead of KRAS lung?

The C6 Gene set enrichment signature indicates cellular pathways/genes, which significantly change after in vitro manipulation of cancer-associated genes. It is mainly used to understand which pathways/genes are dysregulated, independently of the cell type used. For example, in this manuscript (*Haotian et al., The Landscape of Long Non-Coding RNA Dysregulation and Clinical Relevance in Muscle Invasive Bladder Urothelial Carcinoma. Cancers (Basel)*. 2019 Dec 2;11(12):1919.doi: 10.3390/cancers11121919) the authors found that specific lncRNAs expressed in Invasive Bladder Urothelial Carcinoma regulate oncogenic signatures including KRAS LUNG BREAST UP V1 UP, KRAS AMP LUNG UP V1UP, indicating that the lncRNAs identified may be important in the KRAS signalling in Urothelial Carcinoma. The same is true for this paper, (Muzumdar MD et al., *Nat. Commun.* 2017 Oct 23;8(1):1090.doi: 10.1038/s41467-017-00942-5), where the authors report lung and breast signatures (Fig. 9c and Suppl. Data 14) although the experiments have been performed in pancreatic cancer cells.

8. According to the hypothesis, E2F, KRAS and EMT should be seen in common among *KIMAT1* KD, *DHX9* KD and *NPM1* KD. However, only KRAS kidney showed the three KDs in common. KRAS lung only shows a downregulation when knockdown *NPM1*. Also, KRAS downstream target RAF and MEK do not have three KDs in common either. The authors should explain this.

- We have regenerated figure 5f ($p < 0.05$) showing all the pathways in common between *KIMAT1*, *DHX9* and *NPM1*. The MEK and RAF pathways are indeed in common between all three conditions and also EMT (Supplementary Fig. 10a).
- It is not surprising that the pathways in common are not exactly the same, this matches the Venn diagrams in Fig. 5d:

Number of genes regulated by *KIMAT1* = 6278

Number of genes regulated by *DHX9* = 1133

Number of genes regulated by *NPM1* = 1324

(*KIMAT1* and *DHX9* common genes) = $477/6278 = 7.6\%$

(*KIMAT1* and *NPM1* common genes) = $581/6278 = 9.25\%$

(*DHX9* and *NPM1* common genes)/*DHX9* = $587/1133 = 51.81\%$

(*DHX9* and *NPM1* common genes)/*NPM1* = $587/1324 = 44.34\%$

This suggests that only about 20% of *KIMAT1*-regulated genes may be controlled by *DHX9* and *NPM1*, while *DHX9* and *NPM1* have almost 50% of common regulated genes.

9. *DHX9* and *NPM1* overexpression models may need to be included during analysis

or performing qRT-PCR showing the changing of KRAS target genes after overexpressing DHX9 and NPM1.

- This experiment has been added in Supplementary Fig.9c.

There is one figure in Supplementary Figure 5 showed the EMT marker but only one marker. There are many EMT markers and KRAS downstream targets (e.g., RAF, MEK and ERK) need to be detected.

- These experiments have been added in Supplementary Fig.10b and 10c.

10. Can these markers be added into the measurements when demonstrating KIMAT1 and its interacting proteins regulate EMT and KRAS signaling?

- As suggested by the reviewer, we have analysed by qPCR and western blot more EMT markers and KRAS targets regulated by KIMAT1, DHX9 and NPM1. The results are reported in Supplementary Fig.9b-d and Supplementary Fig.10b and 10c.

11. From Figure 6f, the western blot apparently showed that KIMAT1 not only affects the stability of DHX9 and DDX5 but also interaction among them. Moreover, Figure 6d showed the binding between DHX9 and NPM1, is this interaction mediated by KIMAT1 or these two proteins in the nucleus can interact with each other? The role of KIMAT1 in DHX9, NPM1, DDX5 and the interaction among these three proteins need to be demonstrated/discussed in more detail.

- As suggested by the reviewer, we carried out DHX9/NPM1 immunoprecipitation in presence of RNase A and, surprisingly, the interaction between DHX9 and NPM1 was still detected (Fig. 6f). This suggests that DHX9 and NPM1 interact independently of KIMAT1 in the nucleus, whilst KIMAT1 is important for the interaction between DHX9 and NPM1 with DDX5. At page 12 we modified the text which now reads: "The association of DHX9 and NPM1 with DDX5 was abrogated by treatment with RNase A, indicating that RNA molecules, including *KIMAT1*, may be important for the binding of DHX9 and NPM1 to the MC (Fig.6f). However, we still detected a binding between DHX9 and NPM1 in presence of RNase A. Thus, DHX9 and NPM1 may interact in the nucleus independently of *KIMAT1* (Fig. 6f)".

12. Supplementary Figure 6d and f showed DHX9 and NPM1 do not directly bind to miR-200, 27, 7, 139 but can still regulate these miRNAs processing, which showed in Supplementary Figure 6b, c. What could be the potential mechanism to explain this? Or is this due to the decreasing of p21 expression? Then how KIMAT1, DHX9, and NPM1 regulate p21 expression?

- Yes, the reviewer is right. DHX9 and NPM1 can still regulate tumor suppressor (TS) microRNAs by modulating p21 expression (Supplementary Fig.12i). This occurs because KIMAT1, DHX9 and NPM1 regulate MYC expression, a well-know p21 repressor (Figure 7a, Supplementary Fig.12a-c and 12i). MYC is a potential target of these TS microRNAs as evidenced by network analysis (Supplementary Fig.12d).

13. Figure 7e, f showed p21 affects the binding between DHX9, DDX5, and the binding between DDX5 and NPM1. At the same time, p21 overexpression downregulated the protein expression of these three proteins by post-transcriptional regulation. What could be the potential explanation or mechanism? How p21 regulate these three proteins expression level and how KIMAT1 regulates p21?

- We have shown that KIMAT1 silencing decreases MYC expression, accordingly inducing p21 (Fig. 7a). p21 increased expression is MYC-dependent (Suppl. Fig. 12a-c). p21 OE reduces DDX5 and NPM1 levels and this effect is post-transcriptional. Indeed, as shown in Suppl. Fig. 12h, DDX5 and NPM1 mRNA is not affected by p21 OE. This suggests that p21 regulates DDX5 and NPM1 post-transcriptionally possibly by controlling the processing of specific microRNAs targeting DDX5 and NPM1. Further investigation on which specific microRNAs are regulated by p21 and target DDX5 and NPM1 is, however, out of the scope of this manuscript.

14. Figure 8c showed that KIMAT1 KD led to increasing in KRAS. This is the opposite of the mechanism and what the text was described. Why?

- We apologize, as the wrong graph has been included in this figure. KRAS correct quantification has now been added in Suppl. Fig.14b. We thank the reviewer for spotting this mistake.

15. KIMAT1 KD mouse model should be included to addressing the metastasis ability.

- We agree with the reviewer, however we could not perform this experiment because cells upon KIMAT1 KD show massive cell death about 72h after GpRs transfection. Therefore, our concern was that the absence of metastases upon KIMAT1 KD could be due to the massive cell death. For this reason, we decided to overexpress KIMAT1 in cancer cells to verify KIMAT1's capacity to give rise to distant metastases (Fig. 8c, d, Supplementary Fig. 15 a-d).

16. Authors should demonstrate in the mouse model that if DHX9 or NPM1 KO can abolish the metastasis function of KIMAT1 in KIMAT1 overexpression cells.

- As suggested by the reviewer we performed this experiment, which is reported in Supplementary Fig.17a-d. DHX9 KO or NPM1 KO partially abrogated KIMAT1 metastatic potential.

Minor Concerns:

1. Figure legends such as Figure 4d, 4j is not clear. The labeling of the figures such as Figure 3d, 3f, the cell line labeling is confusing.

- We addressed this point.

2. It is better if the authors could include the Xenograft model of KIMAT1 KD by treating LNA-GpR in Figure 3.

- We could not include the Xenograft experiment in Fig. 3 due to lack of space.

3. In Figure 4h, j, the IP efficiency should be shown by western blot.

- This experiment has been added in Fig. 4h.

4. In several qRT-PCR results, the statistical significance labels are not clear. There was more than one comparison between bars, but there is only one significance label showing in the figures. The authors should fix this problem.

- We revised the statistics in the following figures: Fig. 2g, 3c, 3d, 3f, 5g, 5h, 6b, 7b, 7d, Supplementary Figure 1i, 1m, 4a, 4d, 4f, 4g, 6d, 6g, 6i, 11b, 11c, 11d, 11f, 12b, 13d.

5. Western Blot in Figure 6c, e should show a longer time exposure blot.

- We have addressed this point.

6. In Figure 7b, statistical significance could be shown.

- We added statistical significance in Fig. 7b.

7. In Figure 8f, mice bearing H460 cells, which is a KRAS Mut metastatic cell line. However, after the growth of 4 weeks, neither the EV treatment or the KIMAT1 treatment showed metastasis to other organ such as liver or kidney. Why?

In this experiment mice were sacrificed when they started to be unwell. Histology showed that 100% of the mice injected with cells overexpressing KIMAT1 had micrometastases while only 25% of the mice injected with cells overexpressing an empty vector had micrometastases in the liver (Supplementary Fig. 15d). In the text we have made it clear that we are talking about micrometastases rather than metastases in this specific experiment.

Reviewer #2 (Remarks to the Author): Expert in lncRNA

Shi and colleagues describe a new KRAS-responsive lncRNA that has important roles in driving tumorigenesis. A model is proposed in which the lncRNA promotes processing of oncogenic miRNAs and halts biogenesis of miRNAs with tumor suppressor function. There is a lot of data in the paper and it is generally clear. It should be of broad interest. The authors should address the following points:

(1) The authors have used CAGE to map the 5' end of the lncRNA, but it does not appear they have validated the exact sequence of the mature transcript. Where is the 3' end? Is it spliced as the annotation suggests? This is important to ensure that the overexpression plasmids are generating the physiologically relevant transcript. The authors should also mention if it is conserved in any other species.

- KIMAT1 (AP001065.15 or lncRNA02575, ENSG00000228709) has been annotated well. The 5' end, 3' end and full sequence are available from the Ensembl database. There is only one KIMAT1 isoform and it is not conserved in other species. As requested by the reviewer, we added this information in Supplementary Fig. 1e and 1f.

(2) Fig 3: It would be really nice to show that the authors can rescue some of the gapmer effects with a lncRNA expression plasmid that is resistant to the LNA-gapmer. This would prove without any doubt that the effects are “on-target”.

- As suggested by the reviewer, we performed a rescue experiment. Specifically, we overexpressed KIMAT1 with or without the sequence targeted by the GapmeRs (KIMAT1 WT or KIMAT1 mut) in H1299 and H460 cells and performed functional assay, including colony and 3D cell invasion assay. Results are reported in Supplementary Fig. 5d-f.

(3) Fig 6b: Can the authors speculate on why the mature and precursor data do not correlate very strongly in many cases? For example, for 200a, the precursor goes up 5 fold with gapmer #1, but the mature is only slightly increased.

- The primers, protocol and programme of qPCR used to analyse mature and precursor miRNAs are different. Also house keeping genes for mature and precursor miRNAs are different. Therefore, it is not surprising to see different fold changes between mature and precursor microRNAs.

(4) It would be helpful in the discussion to talk about how the authors envision DHX9/NPM1 only regulates a subset of pri-miRNAs.

- As shown in Suppl. Fig. 11d DHX9 and NPM1 bind only to specific miRNAs with oncogenic function. p21 binds to TS miRNAs (Suppl. Fig. 12g). p21 OE reduces the processing of oncogenic microRNAs by hampering the interaction between DDX5 and DHX9 and between DDX5 and NPM1 (Fig. 7d-f). We have reported this in the discussion, which reads: “*KIMAT1*, through DHX9 and NPM1 stabilization and MYC-dependent suppression of p21, promotes the processing of a subset of oncogenic-like miRNAs while simultaneously halting the biogenesis of a subset of miRNAs with tumor suppressor function, maintaining a positive feedback loop that potentiates the KRAS signaling (Fig. 8h). To our knowledge, this is the first study that reports an antagonistic effect on miRNA processing by members of the MC to promote or suppress tumorigenesis. Further investigation would be fundamental in defining the effective number of pri-miRNAs that bind to DHX9/NPM1 or p21 and how the recognition occurs”.

(5) Drosha processing occurs in the nucleus but most of the lncRNA is in the cytoplasm, so there is a bit of a disconnect here that the authors should discuss more in the text.

- As shown in Fig. 3a,b KIMAT1 is present in both cytoplasm and nucleus, although to a lesser extent in the nucleus (~40%) compared to the cytoplasm (~60%). The fact that DHX9 and NPM1 are also present in both cytoplasm and nucleus further corroborates the fact that KIMAT1 is important for DHX9/NPM1 functions in these compartments. This point has been added in the discussion at page 16: “By RAP-MS and CLIP assay we discovered that *KIMAT1* binds to and stabilizes DHX9 and NPM1, which are present in both the nucleus and cytoplasm^{22, 50}, in accordance with *KIMAT1* localization”.

Minor points

(1) P.3: Some words are missing in the sentence that begins “MiRNAs bind to the

3'UTR of target genes.” I believe the authors meant to say that miRNAs can induce mRNA degradation.

- We apologize for this mistake, part of the sentence was somehow deleted while formatting the manuscript. We have now addressed this point.

(2) I see that a few papers refer to the Microprocessor complex as MC, but this does not appear to be very standard in the field.

- The reviewer is right, it is not common to refer to Microprocessor Complex as MC. We used this abbreviation (MC) only to simplify the text and the model reported in Fig. 8f.

(3) P.4: For a general audience, please explain what the difference between “amplification of the KRAS gene” and “copy number gain of the KRAS gene.” They sound like the same thing to me.

- This point has been addressed in the introduction at page 4, which now reads: “Through *in silico* analysis of KRAS copy number alteration (CNA) in human clinical samples from the Cancer Genome Atlas (TCGA), we identified amplification (CN=2 or more) of the *KRAS* gene, as previously reported., as well KRAS copy number gain (CN=1) in both lung adenocarcinoma (LUAD) and lung squamous cell carcinoma (LUSC), with consequent increase of *KRAS* mRNA (Fig. 1a and Supplementary Fig. 1a)”.
- In addition, difference between deep deletion, shallow deletion, diploid, gain and amplification has been better specified in the figure legend (Fig.1a).

(4) P.5: Supp Fig 1h is called out in the wrong place in the text.

- This point has been addressed.

(5) Supp Fig 1l: Please include EV and KRAS labels on x-axis.

- This point has been addressed. Now it is Supplementary Figure 2b.

(6) Fig 2d: The x-axis could be labeled more simply.

- We addressed this point.

(7) Supp Fig 4h: It looks like GAPDH levels also increase so this is not a good control. Quantification of blots in Supp Fig 4h-k would be helpful.

- We added the quantification in Supplementary Figure 7a-c.

(8) P.9: “To define the transcriptome modulated by KIMAT1 and its interacting proteins we used RNA-seq”: RNA-seq does not reveal interacting proteins.

- We have rephrased this sentence which now reads: “To define the transcriptome modulated by KIMAT1, DHX9 and NPM1 we used RNA-seq in H1299 cells

transfected with either *KIMAT1*-targeting GpRs or a pool of four different siRNAs targeting DHX9 or NPM1” at page 10 in the text.

(9) Fig 5d: It would be more helpful to have all 3 datasets together rather than separated out as currently displayed.

- We have replaced the Venn diagrams in Fig.5d.

(10) Fig 6c: It would be good to use a different miRNA that is not affected in cells by KIMAT1 KD and show that its in vitro processing is not altered.

- We performed this experiment in another laboratory, as we do not have permission to use radioactive in our Institute. Due to the COVID-19 restrictions we could not perform this experiment again adding a negative control. However we added a longer exposure of the film, which clearly shows that there is a band in the KIMAT1 KD line, which is exactly the expected size (97 nt) of pre-miR-27b and is missing in the control line.

(11) P.12: The data in Fig 6f is interpreted too strongly in the main text. The RNase A treatment data could support KIMAT1 being involved, but it is also consistent with many other RNAs being involved.

- We agree with the reviewer and addressed this point at page 12. The text now reads: “The association of DHX9 and NPM1 with DDX5 was abrogated by treatment with RNase A, indicating that RNA molecules, including *KIMAT1*, may be important for the binding of DHX9 and NPM1 to the MC (Fig.6f)”.

Reviewer #3 (Remarks to the Author): Expert in mouse models of lung cancer

The authors in this manuscript describe the identification of a KRAS-responsive long non-coding RNA that controls microRNA processing and regulate tumorigenesis. They focused on the Wild-type KRAS (KRASWT) amplification and investigated the role of KRASWT overexpression in lung cancer progression. The authors identified and characterized a KRAS-responsive lncRNA, KIMAT1 and showed that it correlates with KRAS levels both in cell lines and in lung cancer specimens. They also explored the mechanisms to show that KIMAT1 is a MYC target and drives lung tumorigenesis by promoting the processing of oncogenic miRNAs through DHX9 and NPM1 stabilization while halting the biogenesis of miRNAs with tumor suppressor function via MYC-dependent silencing of p21. They conclude that KIMAT1 in maintaining a positive feedback loop that sustains KRAS signaling during lung cancer progression.

The manuscript may be intriguing. However, some of the points and experimental data are confusing especially in some animal studies.

1. The authors used TCGA data to show that high KRAS expression or KRAS amplification had a poorer overall survival (OS) or disease-free survival compared to patients with low KRAS expression. It is confusing that as mentioned that a total of 34% of lung adenocarcinoma (LUAD) and 48.8% of lung squamous cell carcinoma (LUSC), with consequent increase of KRAS mRNA expression (Fig. 1a and Supplementary Fig. 1a). 17% of adenocarcinoma patients with KRAS gain/amplification also harboured a mutant KRAS allele (Supplementary Fig. 1b and Supplementary Table 1).

However, Fig 1b. in LUAD the author used KRAS high vs KRAS low, and in LUSC, the author used KRAS amplified vs not amplified and in amplified group, only 10 cases were included. The reader may be confused why the author use different criteria and what are the rationale? The number of KRAS amplified is also too small to draw any conclusion.

- To avoid confusion, as suggested by the reviewer, we decided to compare for both LUAD and LUSC the survival of patients with KRAS amplification versus patients with diploid KRAS status only (Fig. 1b). In LUSC 10 patients with amplified KRAS were analysed because the survival data in the LUSC dataset from the TCGA are available for these patients only. In LUAD this information is available for 23 patients only. However, we believe that it is remarkable that although the number of patients is low the difference in probability of disease free survival is still significant between patients with or without KRAS amplification.

2. The reader may be interested to know what is the difference in the regulatory mechanism of this pathway between KRAS mutant vs KRASWT with KRAS amplification?

- KRAS activation (either by mutation or amplification) activates the MEK/ERKs pathways and, therefore, induces KIMAT1 expression. However, we showed that in KRAS mutant cells KIMAT1 is expressed at lower levels compared to cells harbouring KRAS amplification (Supplementary Fig. 3e). The higher the expression of KRAS the higher the expression of KIMAT1. This is consistent with the increased expression of KIMAT1 in late stage lung tumors in correlation with KRAS expression (Fig. 1e).
- Regarding the difference in the regulatory mechanism between KRAS WT and mutant cells, we have performed several experiments in KRAS mutant cell lines, including H460 and A549 cells (Fig. 3f, 3h, 7c, 7g, Suppl. 1k, 1m, 5b, 5c, 5f, 6h, 6i, 12a, 12f, 13b, 13d, 13e, 15c-d) which show, as expected, that the mechanism and pathways regulated by KIMAT1 is essentially the same between KRAS mutant and wild type cells.
- Additionally, Suppl. Figs. 1c, d clearly show that both WT and Mutant KRAS activate the same pathways, as reported in the first lines of the discussion at page 16.
- We discussed this point in the text (page 17): "Although *KIMAT1* expression is lower in cells with a KRAS mutational status compared to those with amplified KRAS, the biological effects upon *KIMAT1* manipulation in these cells are similar to those observed in cells with KRAS amplification".

3. The author showed that MYC and regulate KIMT1 expression, but it is not clear whether KRAS regulate KIMT1 is MYC dependent.

- We overexpressed KRAS wild-type or mutant in H1299 cells and simultaneously silenced MYC. When MYC is silenced, KIMAT1 induction is significantly reduced. These results are shown in Fig. 2g.

4. As the author mentioned that KIMAT1 in maintaining a positive feedback loop that

sustains KRAS signaling and KIMAT1 KD induces MYC downregulation. However, in Fig. 8h, the feed back regulatory mechanism was not seen.

- We thank the reviewer for the suggestion. The feedback has now been added to the model in Figure 8f.

5. The animal study in Fig. 8 b,d,e,g and Supplementary 8 c, f, g are confusing, the author used arrow to indicate the tumors, however the readers may not be convinced that they are really metastatic tumors. The resolutions are not adequate, lower power field pictures are not shown to tell the difference of tumor nodules and the arrow heads are too big, even bigger than the xenograft tumor nodules.

- As suggested by the reviewer, we reduced the size of the arrows in Fig. 8c. and added lower magnification images in Fig. 8d, 8e, Suppl. Fig. 15a, 15d, and 16a.

6. In p 15, a patient with limp nodes metastasis, should it be with lymph nodes metastasis?

- Thank you for spotting this typo, we have now addressed this point.

Reviewer #2, expert in ncRNA (Remarks to the Author):

The authors have addressed several of my concerns, but I feel a couple things still should be clarified.

(1) Two reviewers requested the authors to better annotate the gene structure of KIMAT1. In response, the authors have added a note to p.5 that ENSEMBL says there is only one isoform of the lncRNA, but this does not mean that the annotation is correct. The authors should show evidence that the KIMAT1 transcript is spliced and terminated at the 3' end where it is annotated. If the transcript is not spliced or terminated at this location, the overexpression results throughout the manuscript are of unclear significance.

(2) Supp Fig 5e: I appreciate the authors including a rescue experiment but it appears that mutating the GpR targeting site itself caused a significant effect on colony size. This may be interesting as the authors have found a functional element in their lncRNA or it may suggest the overexpression results are fairly noisy.

Writing suggestions:

(1) Page 4: The description of amplification vs. copy number gain may still be confusing to some readers with minimal cancer expertise. I assume that "CN=2 or more", means 2 or more in addition to the 2 copies of the gene that are normally present. This could be better clarified in the text.

(2) Supp Fig 6c,d: On p.9, the authors write that this revealed "that the binding with DHX9 or NPM1 is important for KIMAT1-mediated cell proliferation." Formally, this experiment only shows that the mutated regions are required. This may involve DHX9 or NPM1, but also may involve other factors so more careful language should be used.

Reviewer #3, expert in mouse lung cancer (Remarks to the Author):

The authors had responded adequately to some of the comments. However, the pathological pictures and interpretations especially in Fig.8 are still inadequate and confusing. I strongly suggested a well-experienced expert in animal pathology to look over and verify these data. For example, in Fig. 8d, what does the arrow head in H1299 Ev liver mean? The Fig legend did not mention the meaning of the arrow head.

REVIEWER COMMENTS

Reviewer #2, expert in ncRNA (Remarks to the Author):

The authors have addressed several of my concerns, but I feel a couple things still should be clarified.

(1) Two reviewers requested the authors to better annotate the gene structure of KIMAT1. In response, the authors have added a note to p.5 that ENSEMBL says there is only one isoform of the lncRNA, but this does not mean that the annotation is correct. The authors should show evidence that the KIMAT1 transcript is spliced and terminated at the 3' end where it is annotated. If the transcript is not spliced or terminated at this location, the overexpression results throughout the manuscript are of unclear significance.

- As suggested by the reviewer we have performed rapid amplification of cDNA ends (RACE), which showed that indeed KIMAT1 is spliced at the 3' end and KIMAT1 full length is 912 nt. These results have been reported in Suppl. Fig. 1f. We also confirmed that there is only one isoform of KIMAT1, as evidenced by the single band in lane 1 (Supplementary figure 1f).

(2) Supp Fig 5e: I appreciate the authors including a rescue experiment but it appears that mutating the GpR targeting site itself caused a significant effect on colony size. This may be interesting as the authors have found a functional element in their lncRNA or it may suggest the overexpression results are fairly noisy.

- The reviewer is right. The binding sites for KIMAT1 GapmeR #1 and #3 are located in the region of KIMAT1 required for the interaction with DHX9 or NPM1, respectively (Figure 4g). Indeed, overexpression of KIMAT1 deletion fragments (without DHX9 or NPM1 binding sites) only minimally affects cell proliferation as compared to KIMAT1 full length OE (Supplementary Figure 6c,d), which is consistent with the results in Supplementary Figure 5e,f. We have highlighted this point in the text at page 8.

Writing suggestions:

(1) Page 4: The description of amplification vs. copy number gain may still be confusing to some readers with minimal cancer expertise. I assume that "CN=2 or more", means 2 or more in addition to the 2 copies of the gene that are normally present. This could be better clarified in the text.

- We have reported the TCGA annotation for copy number variation, where gain is low level amplification and indicates at least one KRAS copy more than the two normally present and amplification indicates at least two KRAS copies more than the two normally present. Distinction between the two is not entirely possible. However, in general gain indicates low level amplification while amplification indicates high level

amplification. We have added these details in the method section, specifically in the paragraph titled “CCLE and TCGA KRAS copy number “ and in the legend of Fig. 1a.

(2) Supp Fig 6c,d: On p.9, the authors write that this revealed “that the binding with DHX9 or NPM1 is important for KIMAT1-mediated cell proliferation.” Formally, this experiment only shows that the mutated regions are required. This may involve DHX9 or NPM1, but also may involve other factors so more careful language should be used.

- We thank the reviewer for pointing this out. We rephrased the sentence, which now reads: “Overexpression of the mutants gave rise to a lower number of colonies compared to cells transfected with *KIMAT1* full length (Supplementary Fig. 6c,d), revealing that the regions of *KIMAT1* binding to DHX9 or NPM1 are important for *KIMAT1*-mediated cell proliferation and corroborating previous findings (Supplementary Figure 5e,f)”.

Reviewer #3, expert in mouse lung cancer (Remarks to the Author):

The authors had responded adequately to some of the comments. However, the pathological pictures and interpretations especially in Fig.8 are still inadequate and confusing. I strongly suggested a well-experienced expert in animal pathology to look over and verify these data. For example, in Fig. 8d, what does the arrow head in H1299 Ev liver mean? The Fig legend did not mention the meaning of the arrow head.

- The IHC slides were previously reviewed by two independent pathologists. Nevertheless, we asked a third animal pathologist to review figures and legends. No issues were raised. However, as suggested by the reviewer and to avoid confusion, dashed lines instead of arrows were used in Fig. 8c and more details were added in the legends, specifically in Figs. 8d, 15a, 15d, 17b (highlighted in red in the text).

Reviewer #2 (Remarks to the Author):

The authors have adequately addressed my remaining minor concerns.

Reviewer #3 (Remarks to the Author):

No further comment.